# Socrates Loss: Unifying Confidence Calibration and Classification by Leveraging the Unknown

**Sandra Gómez-Gálvez**[*]                                                    *sgom490@aucklanduni.ac.nz*
*University of Auckland*
**Tobias Olenyi**
*University of Auckland*
*Technical University of Munich*
**Gillian Dobbie**
*University of Auckland*
**Katerina Taškova**[*]                                                *katerina.taskova@auckland.ac.nz*
*University of Auckland*

[*] *corresponding authors*

**Reviewed on OpenReview:** *https://openreview.net/forum?id=DONqw1KhHq*

## Abstract

Deep neural networks, despite their high accuracy, often exhibit poor confidence calibration, limiting their reliability in high-stakes applications. Current ad-hoc confidence calibration methods attempt to fix this during training but face a fundamental trade-off: two-phase training methods achieve strong classification performance at the cost of training instability and poorer confidence calibration, while single-loss methods are stable but underperform in classification. This paper addresses and mitigates this stability-performance trade-off. We propose *Socrates Loss*, a novel, unified loss function that explicitly leverages uncertainty by incorporating an auxiliary *unknown* class, whose predictions directly influence the loss function and a dynamic uncertainty penalty. This unified objective allows the model to be optimized for both classification and confidence calibration simultaneously, without the instability of complex, scheduled losses. We provide theoretical guarantees that our method regularizes the model to prevent miscalibration and overfitting. Across four benchmark datasets and multiple architectures, our comprehensive experiments demonstrate that Socrates Loss consistently improves training stability while achieving more favorable accuracy-calibration trade-off, often converging faster than existing methods.

## 1 Introduction

Deep neural networks (DNNs) have achieved remarkable performance across diverse domains, yet their deployment in high-stakes applications remains constrained by their ability to reliably operate in real-world conditions. Critical applications, such as medical diagnosis (Gireesh & Gurupur, 2023), nuclear security (Ayodeji et al., 2022), and biosecurity (McEwen et al., 2021), require models that provide both accurate, reliable, and trustworthy predictions. One important aspect of a reliable model is its ability to effectively represent its own uncertainty. To achieve this, a model needs to be *calibrated*, i.e., its predictive confidence matches the true likelihood of correctness (Guo et al., 2017).

In classification settings, confidence scores from softmax layers commonly serve as a proxy for uncertainty (Abdar et al., 2021), where, ideally, predictions made with 90% confidence should be correct 90% of the time. However, while modern DNNs are widely known to suffer from systematic overconfidence (i.e., higher confidence than the actual accuracy) (Guo et al., 2017), recent evidence shows that they may also be underconfident or exhibit mixed calibration patterns depending on architectural choices (Minderer et al.,

2021). As we also demonstrate in this work, modifying the loss function alone can produce underconfident or mixed-calibration models. This gap between predicted confidence and actual accuracy undermines the reliability needed for high-stakes applications.

Research in confidence calibration has emerged to address this challenge by both quantifying miscalibration and developing methods to improve the alignment between predictive confidence and accuracy. Current alignment methods broadly fall into two categories: post-hoc methods (Fisch et al., 2022; Galil et al., 2023; Moon et al., 2020; Naeini et al., 2015; Platt, 2000; Kull et al., 2019; Zadrozny & Elkan, 2001; 2002) that adjust confidences after training without modifying model parameters, and ad-hoc methods (Hendrycks et al., 2020; Lakshminarayanan et al., 2017; Lin et al., 2020; Mukhoti et al., 2020; Müller et al., 2019; Pereyra et al., 2017; Thulasidasan et al., 2019) that integrate calibration during training.

While post-hoc methods offer simplicity and speed, they face significant limitations, including hyperparameter tuning and additional data, which is problematic when data is scarce (Bohdal et al., 2023; Kim & Yun, 2022; Wang et al., 2021). We identify a critical logical gap: Many widely used pre-trained models are optimized for accuracy and are not well-calibrated, often lacking available data for post-hoc calibration, limiting their reliability in downstream tasks. Furthermore, post-hoc methods are fundamentally incompatible with knowledge-transfer paradigms such as transfer learning, where calibrated representations must be embedded within the model weights themselves (You et al., 2020). The impact of performing transfer learning with calibrated versus uncalibrated pre-trained models remains unexplored.

In contrast, ad-hoc methods integrate calibration into the training, creating their own challenges (Le Coz et al., 2024), such as longer development times and a trade-off in accuracy to achieve better calibration. Current methods typically act as regularizers using data augmentation (Hendrycks et al., 2020; Thulasidasan et al., 2019), adapted loss functions (Lin et al., 2020; Mukhoti et al., 2020; Müller et al., 2019; Pereyra et al., 2017), or modifying the model architecture (Lakshminarayanan et al., 2017).

Through empirical evaluation, we identify that existing ad-hoc confidence calibration methods face a fundamental trade-off: methods that combine losses in a two-phase training achieve strong classification performance but suffer from training instability and worse calibration, while single-loss methods train stably and achieve strong calibration performance but at the cost of lower classification performance. Reliability diagrams show that final models in both cases are underconfident or overconfident, with calibration fluctuating across epochs and dependent on dataset and architecture. These insights motivate our research question: *Can we design an ad-hoc calibration method based on a single, easily implementable, loss function that trains a single model while promoting training stability and maintaining calibration and classification performance across diverse datasets and architectures?*

In exploring the connection between ad-hoc confidence calibration and ad-hoc selective classification due to their regularization nature (Galil et al., 2023), we analyze Self-Adaptive Training (SAT) (Huang et al., 2020) and uncover a relationship between the average confidence assigned to the unknown class and confidence calibration. While correlation does not imply causation, this observation led us to extend our research question: *Does explicit uncertainty modeling through an unknown class mitigate training instability while maintaining calibration and classification performance?*

To this end, our contributions are threefold:

- We propose **Socrates Loss**[1], a novel ad-hoc and easy-to-implement confidence calibration method that mitigates the training stability-performance trade-off by unifying classification and confidence calibration objectives through explicit uncertainty modeling via an *unknown* class. By incorporating predictions for this class into the loss function via a dynamic uncertainty penalty, Socrates Loss penalizes the model for failing to recognize its own uncertainty. The loss further emphasizes hard-to-classify instances and leverages previous predictions to adaptively promote training convergence and stability.

---

[1]Socrates Loss was named after the philosopher Socrates and his famous quote *I know that I know nothing.*

- We provide theoretical guarantees showing that Socrates Loss regularizes the weights of the network and acts as a regularized upper bound on the Kullback-Leibler divergence, thereby preventing overfitting and miscalibration while maintaining training stability.

- We demonstrate empirically that Socrates Loss improves confidence calibration performance, achieving a better training stability and accuracy–calibration trade-off across multiple benchmarks, architectures, and transfer learning scenarios, without compromising accuracy.

## 2 Background and Related Work

The level of confidence calibration can be assessed both visually and quantitatively (for formal definitions and extended discussion see Appendix B). A widely used visualization method is the reliability diagram (Niculescu-Mizil & Caruana, 2005), which plots the expected sample accuracy as a function of prediction confidence at a given training epoch, following several binning strategies (Filho et al., 2023; Guo et al., 2017; Nguyen & O'Connor, 2015). We adopt the approach in Guo et al. (2017), which groups confidences into $M$ interval bins of size $1/M$. While reliability diagrams offer visual insights, evaluating confidence calibration only at the final epoch is insufficient, particularly for ad-hoc confidence calibration methods that influence training dynamics. Following Lin et al. (2020), we argue that ad-hoc confidence calibration should be assessed across training epochs. To jointly analyze classification and confidence calibration across epochs, we suggest the use of Pareto plots, providing a more holistic visualization of the performance trade-off.

Quantitatively, the most common metric is the Expected Calibration Error (ECE) (Naeini et al., 2015), which measures the average bin-wise discrepancy between confidence and accuracy. To account for potential binning biases and to evaluate calibration at a more granular level, researchers have proposed variants such as AdaptiveECE (AdaECE), which ensures an equal number of samples per bin, and Classwise-ECE (CW-ECE), which extends the ECE calculation across all classes (Mukhoti et al., 2020). While these metrics provide the necessary tools to evaluate calibration, they are not sufficient to evaluate general performance. As noted by Zhang et al. (2023), a well-calibrated model may be a poor discriminator, and vice versa. Therefore, calibration metrics should be interpreted alongside accuracy to evaluate model performance.

The current literature lacks a precise criterion for determining whether a model is ECE confidence-calibrated, as it depends on the risk tolerance of the use case. We define a model acceptably calibrated for less critical tasks below 10%, well-calibrated as close to 0%, and perfect calibration when is 0%.

A common approach to address miscalibration is through post-hoc methods, which adjust the outputs of a pre-trained model without altering its learned parameters. Prominent multi-class examples include Temperature Scaling (TS) (Guo et al., 2017), which recalibrates logits using a single learned parameter; and Matrix Scaling and Vector Scaling (Guo et al., 2017), variants of the well-known binary method Platt Scaling (Platt, 2000). While simple and computationally efficient, post-hoc methods have fundamental limitations, such as confidence degradation in correct predictions (Bohdal et al., 2023), limited efficacy in some settings (Wang et al., 2021; Kim & Yun, 2022), need an additional labeled validation set for tuning calibration hyperparameters, and not applicable before knowledge transfer. For applications like transfer learning, if we want to initiate from a calibrated model (as in You et al. (2020)), calibrated representations must be embedded within the model weights themselves, a requirement that post-hoc methods cannot fulfill.

To overcome these issues, ad-hoc methods integrate calibration directly into the training process. One path to calibrate is through the loss function. The core challenge for these methods lies in modifying the training objective to promote calibration without sacrificing accuracy. Current methods generally fall into two categories: single-loss methods that modify the primary loss function, such as Focal Loss (Lin et al., 2020), Adaptive Sample-Dependent Focal Loss (FLSD) (Ghosh et al., 2022), Meta-Calibration (MC) (Bohdal et al., 2023), or Brier Loss (Mukhoti et al., 2020), and methods that combine losses or use complex training schedules, such as Confidence-aware Contrastive Learning for Selective Classification (CCL-SC) (Wu et al., 2024). However, this has led to a critical trade-off: single-loss methods tend to be stable but often provide limited classification improvement, while methods that combine losses in a two-phase training can achieve stronger classification performance but frequently suffer from implementation complexity, training instability, and poorer confidence calibration, a finding that our experiments corroborate. Another path to calibrate is

changing the architecture. Among the most widely adopted methods is Deep Ensembles (Lakshminarayanan et al., 2017), which demonstrates strong calibration performance but is computationally expensive and prone to increased overfitting (Shashkov et al., 2023).

A related line of work in Selective Classification offers a compelling mechanism for explicitly modeling model uncertainty. These methods allow it to abstain when it is uncertain. This is often achieved by introducing an additional *unknown* or *abstention* class into the model's output layer, as seen in methods like DeepGamblers (Liu et al., 2019) and Self-Adaptive Training (SAT) (Huang et al., 2020). While these two methods have proven effective for selective classification and preventing overfitting, their potential to resolve the ad-hoc calibration trade-off has been underexplored. The use of an unknown class has not yet been leveraged to create a unified and stable optimization objective for confidence calibration.

## 3 Socrates Loss: A Unified Confidence Calibration and Classification Loss

In the pursuit of reliable models, we propose an ad-hoc method to train confidence-calibrated classifiers. To explicitly model uncertainty, we reframe the standard multiclass classification with $c$ classes as a $(c+1)$ classification problem, introducing an additional *unknown* class, denoted as *idk* for mathematical convenience. This allows the model to learn not only what it knows, but also to signal what it does not know. Our method uses information from the ground truth class, while leveraging also information from: 1) an additional unknown class, 2) hard-to-classify instances, and 3) predictions from previous and current epochs. To achieve this, we introduce a novel, easy-to-implement, loss function called *Socrates Loss*, which maintains a unified optimization objective for both classification and confidence calibration.

### 3.1 Formulation

Let $\mathcal{X}$ be the input space, $\mathcal{Y}$ the output space defined by $c+1$ classes. A classifier $f(\cdot)_{c+1} : \mathcal{X} \to \mathcal{Y}$ is optimized by minimizing the Socrates Loss, defined as:

$$\mathcal{L}_{\text{Socrates}}(f) = -\frac{1}{n} \sum_{i=1}^{n} \overbrace{(1-\hat{p}_{i,y_i,e})^\gamma}^{\substack{\text{focal term with} \\ \text{modularity factor}}} \Big[ \underbrace{\overbrace{t_{i,y_i,e}}^{\substack{\text{adaptive} \\ \text{target}}} \log \hat{p}_{i,y_i,e}}_{\text{ground truth component}} + \underbrace{\overbrace{\beta_{i,e}}^{\substack{\text{dynamic} \\ \text{uncertainty penalty}}} \overbrace{(1-t_{i,y_i,e})}^{\substack{\text{adaptive} \\ \text{target}}} \log \hat{p}_{i,idk,e}}_{\text{unknown component}} \Big]; \qquad (1)$$

$$\underbrace{\beta_{i,e}}_{\substack{\text{dynamic} \\ \text{uncertainty penalty}}} = \max_{\bar{y}_i \neq y_i} (\hat{p}_{i,\bar{y}_i,e}) - \hat{p}_{i,idk,e}; \text{ s.t. } \beta \in [0,1]; \qquad (2)$$

$$\underbrace{t_{i,y_i,e}}_{\substack{\text{adaptive} \\ \text{target}}} = \begin{cases} y_i, & \text{if } e \leq E_s. \\ \underbrace{\alpha}_{\substack{\text{momentum} \\ \text{factor}}} \times t_{i,y_i,e-1} + \underbrace{(1-\alpha)}_{\substack{\text{momentum} \\ \text{factor}}} \times \hat{p}_{i,y_i,e}, & \text{otherwise}; \\ & \text{s.t. } \alpha \in (0,1]. \end{cases} \qquad (3)$$

where $\hat{p}_{i,y_i,e}$ and $\hat{p}_{i,idk,e}$ are the predicted probabilities that the i-th instance is associated with the ground truth class $y_i$, and the unknown class $idk$, respectively; $\bar{y}_i$ is any class other than the ground truth class, $n$ is the number of instances, $e$ is the current epoch, $e-1$ is the previous epoch, and $E_s$ is the number of initial epochs before incorporating previous and current predictions to adjust the adaptive target. In the search for end-to-end models, we set $E_s = 0$, where $e = 0$ is the first epoch.

The loss has two hyperparameters: $\gamma$ and $\alpha$. Inspired by Focal Loss, $\gamma$ is a modularity factor within the focal term, that controls the down-weighting of easy instances, i.e., a higher factor gives more weight to difficult instances. Meanwhile, $\alpha$, inspired by SAT, is a momentum factor that adjusts the current target by balancing the influence of previous and current probability predictions, promoting dynamic training convergence and reducing prediction instability.

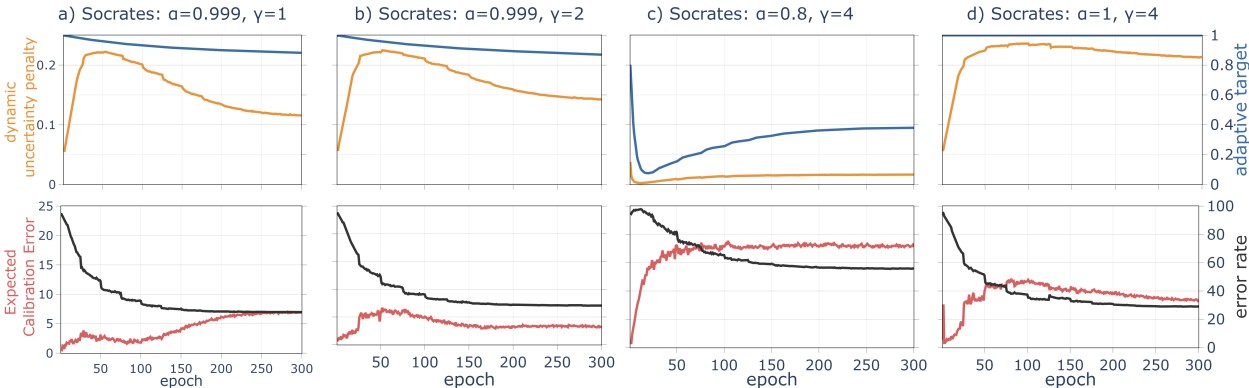

Figure 1: Evolution of sub-components of the Socrates Loss, the dynamic uncertainty penalty (orange curve) and adaptive target (blue curve), alongside the Expected Calibration Error (red curve) and error rate ($1 - accuracy$, black curve) across epochs on CIFAR-100 with VGG-16. Results are averaged over five runs.

The remaining parameter of the loss is the dynamic uncertainty penalty $\beta$, which is not a hyperparameter since it changes dynamically depending on the model's probability predictions. $\beta$ penalizes the model for failing to recognize its own uncertainty, i.e., when any probability not associated with the ground truth class exceeds the probability associated with the unknown class. We exclude the ground truth class when computing $\beta$ to focus the penalty on cases where the model fails to recognize uncertainty. Including it could penalize the model unfairly, while not reflecting its true uncertainty awareness (motivation in Appendix D). We consider this parameter as a standalone component for calibration; further discussion can be found in Section 4 (Experiments and Results).

The pseudocode and a mathematical example are provided in Appendix C and Appendix D, respectively.

## 3.2 Intuition

If any probability, excluding the one associated with the ground truth class, exceeds the one associated with the unknown class, it indicates the model is more confident in an alternative class than in recognizing its own uncertainty. In this case, the dynamic uncertainty penalty assigns importance to the unknown component of the equation. By *recognizing its own uncertainty*, we mean that if the probability of the ground truth class is not the highest, the highest should be that of the unknown class; conversely, if the ground truth class probability is the highest, the unknown class should have the second-highest probability. This mechanism encourages the model to become more confident in recognizing its own uncertainty. In practice, the dynamic uncertainty penalty is higher in the early epochs of training, gradually decreasing thereafter (Fig. 1a and Fig. 1b, top). If the model successfully recognizes its own uncertainty, ($\beta = 0$), the unknown component is omitted, resembling a variant of Focal loss, influenced by the additional unknown class and the adaptive target, that accentuates the impact of hard-to-classify instances.

The adaptive target plays a role in both components by balancing the ground truth target with previous and current predictions, which adaptively helps to converge and stabilize the training. The tuning of the momentum factor ($\alpha$) impacts this balance (Appendix G). For instance, when $\alpha = 1$ (Fig. 1d), the unknown component is also omitted, resulting in a variant influenced by the unknown class that also resembles Focal Loss. The momentum factor must be tuned to control the emphasis on the unknown component and *idk* predictions (Fig. 1b vs Fig. 1c vs Fig. 1d).

Depending on the dataset and architecture, the tuning process (Appendix G) must adjust the emphasis on hard-to-classify instances through the modularity factor (Fig. 1a vs Fig. 1b). The focal term controls the dominance of easy-to-classify instances by down-weighting their contribution to the loss, thereby shifting the effective gradient focus toward hard-to-classify instances. In the absence of this term, easy-to-classify instances contribute to the loss and gradient even at high confidence levels, increasing ground truth logits and inflating classification margins, i.e., increasing the separation between the ground truth confidence and

competing class confidences. While this margin inflation may improve accuracy, it induces overconfident predictions, as there is no mechanism to regulate logit growth once high confidence is reached, widening the accuracy–calibration gap. By reducing the influence of easy instances, the focal term stabilizes margin growth and implicitly constrains logit scaling, encouraging predicted confidence to better align with instance difficulty.

The additional unknown class acts as an explicit uncertainty mechanism. As detailed in Appendix H, the average confidence for both the ground truth and unknown classes increases over training epochs, with the latter increase due to the dynamic uncertainty penalty, which penalizes the model for failing to recognize its own uncertainty. Correct predictions correspond to higher ground truth confidence, whereas the unknown class exhibits higher confidence when the model fails. Moreover, the frequency of top-1 predictions for the unknown class gradually increases over epochs, demonstrating its dual role as an uncertainty mechanism and as a contributor to regularization.

Overall, the dynamic uncertainty penalty enables the model to recognize its own uncertainty, the adaptive target dynamically promotes convergence and stabilizes training, the focal term adjusts the emphasis on hard-to-classify instances, and the unknown class acts as an uncertainty mechanism. Fig. 1b illustrates a well-calibrated classifier that achieves competitive accuracy.

### 3.3 Theoretical Analysis

In this section, we establish the theoretical foundations of the proposed Socrates method, focusing on two key aspects: its role as a weight regularizer and its formulation as a regularized upper bound on the Kullback–Leibler divergence. These properties explain how Socrates Loss mitigates overfitting, improves calibration, and enhances stability and generalization.

#### 3.3.1 Socrates Loss Regularizes the Weights of the Network.

Guo et al. (2017) and Lin et al. (2020) proved there is a relationship between miscalibration and overfitting (but not the opposite). This occurs when the loss function attempts to further reduce its value even after perfect high confidence has been achieved. Lin et al. (2020) demonstrated that, under the cross-entropy loss (CE), DNNs tend to progressively increase their confidence in incorrect predictions for misclassified instances. In contrast, Socrates Loss introduces a regularization effect by dynamically increasing the penalty on the unknown class in the presence of overfitting. Furthermore, the norms of the weights, $w$, are higher at the beginning of the training compared to those trained with CE. It is when the model starts being miscalibrated that there is a change in the ordering of the weight norms, due to a big increase in the weight norm of the models with CE. This behaviour shows that Socrates Loss acts as a regularizer when the model is sufficiently confident, avoiding miscalibration and overfitting.

Formally , let $\mathcal{L}_{\mathrm{CE}}(f)$ be CE loss, and $\mathcal{L}_{\mathrm{Soc}}(f)$ be Socrates Loss. The gradients of the neural network trained with $\mathcal{L}_{\mathrm{Soc}}(f)$ are smaller than the ones trained with $\mathcal{L}_{\mathrm{CE}}(f)$ when a perfect confidence is reached and the model could start overfitting and become miscalibrated, i.e.,

$$||\frac{\partial \mathcal{L}_{\mathrm{Soc}}(f)}{\partial w}|| \leq ||\frac{\partial \mathcal{L}_{\mathrm{CE}}(f)}{\partial w}||. \tag{4}$$

The proof can be found in Appendix E.1.

#### 3.3.2 Socrates Loss Forms a Regularized Upper Bound on the Kullback-Leibler Divergence.

It is well-known that cross-entropy loss (CE) minimizes (provides an upper bound for) the Kullback-Leibler (KL) divergence between the predicted distribution $\hat{p}$ and the target distribution $q$ over classes, i.e., $\mathcal{L}_{\mathrm{CE}}(f) \geq KL(q||\hat{p})$. KL divergence quantifies the information difference between two distributions. In our case, Socrates Loss minimizes KL divergence while regularizing by increasing the entropy of the predicted distribution and leveraging the predictions associated with the unknown class. The regularization parameters are $\gamma, \beta$, and $\triangle_{reg}$; where $\triangle_{reg} = (1 - t_y)[\gamma \hat{p}_y \log \hat{p}_{idk} - \log \hat{p}_{idk}]$. Therefore:

$$\mathcal{L}_{\mathrm{Soc}}(f) \geq KL(q||\hat{p}) - \gamma \mathbb{H}[\hat{p}] + \beta \triangle_{reg}. \tag{5}$$

$\triangle_{reg}$ is considered a regularization term, as it is derived from a different distribution, the unknown distribution, rather than the ground truth distribution; and $\mathbb{H}[\hat{p}]$ is the entropy of the predicted distribution. Therefore, this regularized entropy increase, along with the regularization applied through the prediction associated with the unknown class, prevents the model from becoming overconfident (Pereyra et al., 2017). Then, substituting the CE with Socrates Loss incorporates a maximum-entropy regularizer to the KL minimization objective. As demonstrated by Lin et al. (2020), higher entropy can prevent overconfident predictions, improving model calibration. Therefore, Socrates Loss forms a regularized upper bound on the KL divergence, avoiding overconfident predictions and improving calibration. The proof can be found in Appendix E.2.

## 4 Experiments and Results

To validate our proposed method, we conduct a comprehensive set of experiments designed to assess training stability, calibration performance, and overall effectiveness against existing methods. We extended the publicly available SAT implementation (Huang et al., 2020) to create a unified framework[2] for hyperparameter exploration, training, and evaluation. Additional model reproducibility details can be found in Appendix F.

### 4.1 Experiment Settings

In this section, we describe the datasets and architectures used to validate our method, the baselines methods for comparison, the hyperparameter selection process, implementation details, and the evaluation protocol.

#### 4.1.1 Datasets and Architectures

We evaluate all methods on four benchmark datasets of varying complexity: Street View House Number (SVHN) (Netzer et al., 2011), CIFAR-10/CIFAR-100 (Krizhevsky, 2009), and the large-scale Food-101 (Bossard et al., 2014). These datasets range from simple classification tasks (CIFAR-10) to more challenging real-world scenarios (Food-101), allowing us to test the robustness to task, generalization, and reliability of each method. Although improvements may be less pronounced with the SVHN and CIFAR-10 *toy* datasets, the limitations of the methods could become noticeable. When advanced methods are applied to these datasets, they often introduce unnecessary complexity, highlighting their inefficiency or overfitting tendencies. In line with prior research, we use VGG-16 (Simonyan & Zisserman, 2015) for SVHN and CIFAR-10, and ResNet-34 (He et al., 2015) for Food-101. To assess architectural invariance, we test on CIFAR-100 with three distinct architectures: VGG-16, ResNet-110 (He et al., 2015), and ViT (Dosovitskiy et al., 2021). Since CIFAR-100 lacks sufficient data to train a ViT effectively (Dosovitskiy et al., 2021), we also evaluate CIFAR-100 using a fine-tuned ViT trained through Transfer Learning by replacing the classification head, with no layers frozen during fine-tuning. The initial ViT is a ViT model pre-trained on ImageNet-21K and fine-tuned on Imagenet2012 (*vit-base-patch16-224*, Hugging Face Transformers library (Wolf et al., 2020)).

#### 4.1.2 Baselines

We compare our single-loss ad-hoc Socrates Loss method with: the post-hoc methods Temperature Scaling (TS) (Guo et al., 2017), Matrix and Vector Scaling (MS and VS) (Guo et al., 2017); single-loss ad-hoc methods including cross-entropy Loss (CE), Brier Loss (Mukhoti et al., 2020), Focal Loss (Lin et al., 2020), Adaptive Sample-Dependent Focal Loss (FLSD) (Ghosh et al., 2022), Meta-Calibration (MC) (Bohdal et al., 2023); and methods that combine losses in a two-phase training, such as Confidence-aware Contrastive Learning for Selective Classification (CCL-SC) (Wu et al., 2024) and Self-Adaptive Training (SAT) (Huang et al., 2020). Although SAT was originally proposed as a regularizer to prevent overfitting, its mechanism of using an auxiliary unknown class and an adaptive loss makes it a relevant, albeit unexplored, baseline for ad-hoc calibration.

---

[2]The code is publicly available at https://github.com/sandruskyi/SocratesLoss

### 4.1.3 Hyperparameters and Implementation

We tuned the hyperparameters using the full training and validation sets. For Food-101, the training set was randomly split 80/20. For Socrates, we tested $\gamma \in \{1, 2, 3, 4\}$ and $\alpha \in \{0.8, 0.9, 0.99, 0.999\}$. For the other baselines, we used the hyperparameter values from the original studies. In the absence of such details, we applied the same hyperparameter search, using the ranges provided by the authors. All models were trained for 300 epochs, except for the transferred ViT model, which was fine-tuned for 50 epochs. Results were averaged over five runs (with random seeds 1-5). Full implementation details and hyperparameter settings are provided in Appendix F.

### 4.1.4 Evaluation Protocol

We assess performance using classification accuracy and its error rate $(1 - accuracy)$, and standard calibration metrics: ECE, AdaECE, and CW-ECE (see Appendix B). However, these metrics can be misleading in isolation, as a model may be well-calibrated but inaccurate and vice versa (Zhang et al., 2023). While combined metrics like the Brier score exist, they are often dominated by the accuracy term (Hernández-Orallo et al., 2012). Designing a single metric combining calibration and classification performance remains challenging and represents an open gap. Therefore, to facilitate a more balanced comparison beyond quantitative analysis, we visually assess model performance using Pareto plots, reliability diagrams, learning curves, and metric trends over epochs. We track all metrics, including accuracy, calibration measures, and loss trends, throughout all the training epochs to evaluate stability and convergence dynamics. We select the best model using Pareto plots (error rate versus ECE), resolving ties (i.e., when two models share the same Pareto position) with Classwise-ECE.

## 4.2 Comparative Performance Analysis

We now present the core findings of our experiments, analyzing Socrates method against well-established methods. We focus on four key aspects: training stability, the dynamic accuracy-calibration trade-off during training, transfer learning, and the last-epoch performance at convergence.

**Robust Convergence and Training Stability.** Models trained using Socrates, Focal, FLSD, CE, and Brier methods successfully converged across all datasets and architectures. In contrast, models using SAT on Food-101, and CCL-SC and MC on SVHN (which highlights the dataset challenges) converged prematurely with low accuracy (see Table 1 and Table 2 for test performance and Appendix I Table 9 for validation performance). These issues persisted despite hyperparameter tuning within the recommended ranges and were not raised/addressed in the original studies.

Loss trends (Fig. 2) of the single-loss ad-hoc methods (Socrates, Focal, FLSD, CE, Brier, and MC) consistently reduce the training and validation losses, except for the Food-101 dataset, where Focal and FLSD exhibit an increase in training and validation losses during the final epochs. Furthermore, MC, FLSD and Focal illustrate a nearly flat loss trend (underfitting) for SVHN on VGG-16. These single-loss methods do not show signs of overfitting, suggesting good generalization and potential for calibration. In contrast, SAT and CCL-SC exhibit spikes at the epoch of **combining losses** across all datasets-architectures, as well as overfitting for CIFAR-10 on VGG-16 and CIFAR-100 on ViT when training with SAT. These spikes may contribute to training instability, and these fluctuations coincide with ECE instability observed in both methods (Fig. 3).

For CCL-SC on CIFAR-10, SVHN, and CIFAR-100 using VGG-16, the training loss is higher than the validation loss. This behavior can be expected, as the loss function introduces additional regularization terms active only during training. These terms increase the training loss but improve generalization, which can result in a lower validation loss.

**Dynamic Accuracy-Calibration Trade-off.** Beyond training stability, an effective method must simultaneously improve accuracy and calibration throughout the training process. Figure 3 illustrates this dynamic trade-off by plotting the error rate against ECE over training epochs, where progress towards the bottom-left corner indicates better performance. Models trained with Socrates method consistently follow

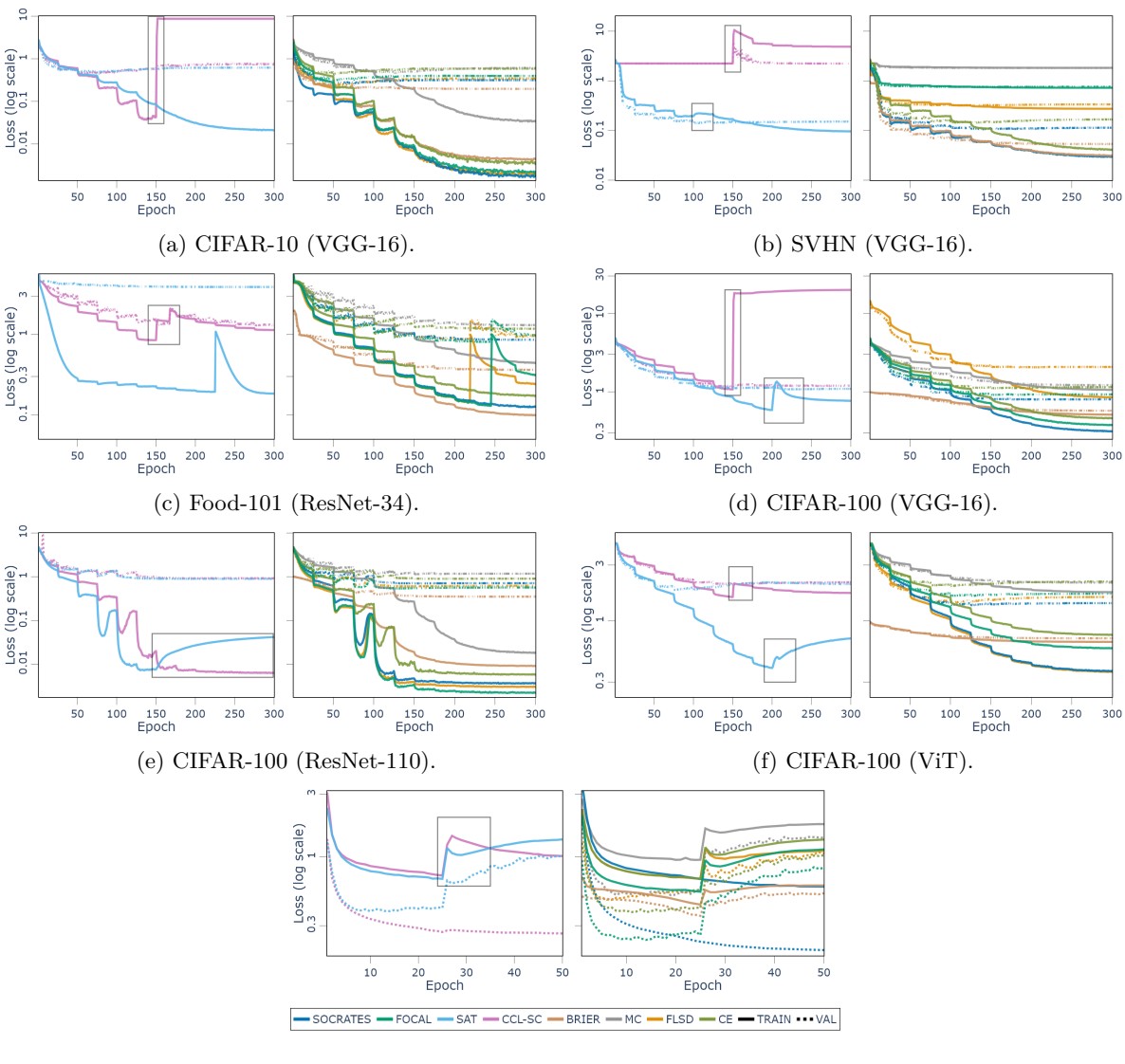

Figure 2: Loss trends for CIFAR-10 with VGG-16 (a), SVHN with VGG-16 (b), Food-101 with ResNet-34 (c), CIFAR-100 with VGG-16 (d), ResNet-110 (e), ViT (f), and ViT Transfer Learning (g). The black rectangles indicate the periods of loss change in the two-phase training approaches.

a direct and efficient trajectory, demonstrating a strong and steady improvement on both axes. Notably, for CIFAR-100, Socrates produces a slight increase in ECE during the initial epochs when accuracy is low; however, ECE begins to improve well before epoch 150. Overall, even when not ranked first (SVHN case), Socrates remains among the top three (e.g., at epochs 150 and 300).

This consistent improvement behavior contrasts sharply with competitors (Fig. 3); for instance, while SAT and Brier improve ECE and error rate on SVHN, they fail to do so on CIFAR-100 with VGG. Models using MC consistently underperform in both calibration and classification. Models trained with CE, despite competitive accuracy, show worsening confidence calibration over epochs, as indicated by increasing ECE values. In the cases where this does not occur (Food-101 with ResNet-34 and CIFAR-100 with ResNet-110), improvements in confidence calibration are marginal. Models with Focal and CCL-SC (excluding premature convergence) generally lag behind the ones using Socrates and FLSD, except for Food-101, where Focal achieves better calibration results than FLSD and CCL-SC. FLSD, CCL-SC, and Focal also show

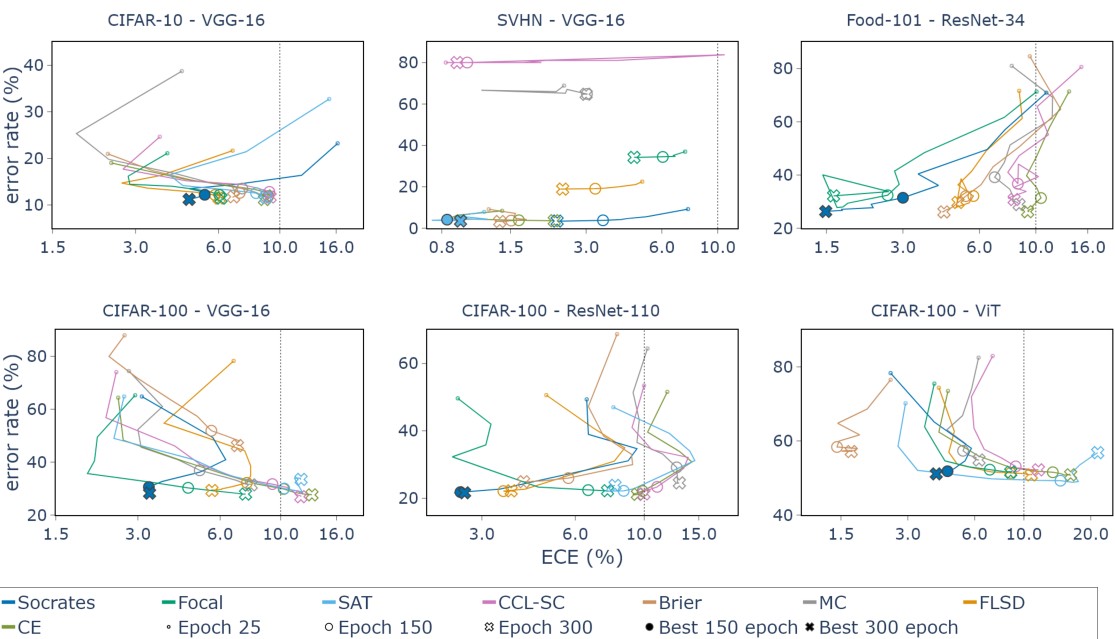

Figure 3: Error rate (1 - accuracy) versus Expected Calibration Error (ECE) across epochs for different datasets-architectures. Dotted lines indicate the threshold for acceptable calibration. Lower and leftward values indicate better performance. SAT is excluded from Food-101 due to premature convergence. Lines are drawn every 25 epochs.

oscillations in ECE and accuracy during the final epochs of Food-101, which are directly connected with the loss spikes illustrated in Fig. 2c). Comparing the models trained with Socrates and FLSD, both methods show overall improvement, but the ones with FLSD suffer from greater ECE instability, decreasing and increasing it; e.g., CIFAR-100 on VGG or Food-101. Several methods (excluding the ones that premature converged) result in models that exceed the acceptable calibration threshold (CE, SAT, and CCL-SC with CIFAR-100 on VGG-16, CCL-SC and MC with CIFAR-100 on ResNet-110, and FLSD, CCL-SC, CE, and SAT with CIFAR-100 on ViT).

For ViT models, Socrates is the only method able to reduce ECE mid-training and prevent severe ECE degradation from start to end as well as reach competitive accuracy (despite CIFAR-100 is not well-suited for ViT without Transfer Learning).

Notably, Socrates method is not only effective but also efficient, often reaching an improved accuracy–calibration trade-off region faster than other methods, as indicated by the epoch markers (e.g., 150 epoch), making it a more reliable calibration and classification method throughout training.

**Transfer Learning.** When training a ViT using Transfer Learning (Fig. 4a), only Socrates and CCL-SC improve both classification and calibration. While CCL-SC shows calibration oscillations in the final epochs, Socrates maintains a stable trend, suggesting that extended training could further enhance both classification and confidence calibration performance, highlighting potential for future research. The fluctuations and performance degradation observed in the other methods indicate an area for further investigation.

At the final epoch (50), the reliability diagram (Fig. 4b) shows that Socrates and CCL-SC methods result in underconfident models, whereas the other methods approximate the perfectly calibrated line more closely; however, this closeness does not translate into classification performance, as their low final accuracy makes them unsuitable as reliable models (see Table 1 and Table 2, bottom group).

These results indicate that Socrates is a promising method for Transfer Learning scenarios, with the potential to further improve performance during extended training.

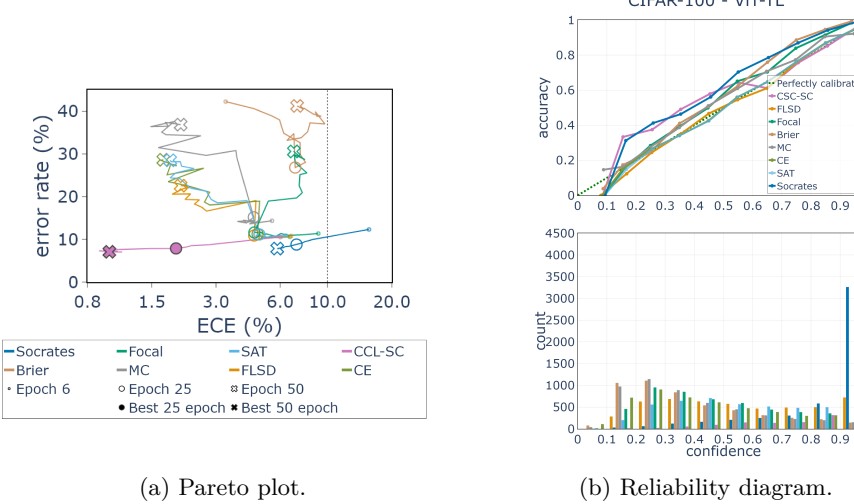

(a) Pareto plot.

(b) Reliability diagram.

Figure 4: Pareto plot across epochs (lines are drawn every 2 epochs) (a) and reliability diagram at the final epoch 50 (b) for the CIFAR-100 validation set using ViT with Transfer Learning (TL). In (b), the SAT curve overlaps with that of CE.

**Last Epoch Performance.** The favorable training dynamics of Socrates method translate to better last-epoch performance than existing confidence calibration methods. Before evaluating the final performance on the test set, it is interesting to analyze the last epoch performance on validation set. The reliability diagram (Fig. 5), along with the previous analysis, illustrates the calibration capacity of Socrates, showing that models trained with Socrates are the closest to the perfect calibration line across all the datasets-architectures. Apart from that, Socrates does not produce overconfident models (except for CIFAR-10), unlike other methods such as SAT or CE. This finding challenges the common assertion that modern Deep Neural Networks systematically suffer from overconfidence (Guo et al., 2017), demonstrating that the choice of loss function can fundamentally alter this behavior. For instance, in CIFAR-100 on ResNet-110, Socrates exhibits a reliability curve close to the perfectly calibrated line, while Focal shows underconfidence. This observation has important implications for the choice of post-hoc confidence calibration methods. Methods explicitly designed to correct overconfident predictions, such as Temperature Scaling, are not well suited for these models, as we discuss below.

Table 1 and Table 2 (CE results) summarize the results at epoch 300 for the test set (see Figure 12 and Figure 14a in Appendix J for visualizing error rate versus ECE as Pareto plot), where Socrates method achieves the best or second-best model (based on its position in the Pareto plot) across the vast majority of dataset-architecture combinations. These findings confirm its ability to strike a balance between high accuracy and strong calibration. For instance, on the challenging CIFAR-100 on VGG-16, Socrates method achieves the lowest ECE (3.45) as well as competitive accuracy (71.26), whereas competitors are forced into a trade-off, achieving either high accuracy for worse calibration (e.g., CCL-SC) or vice-versa (e.g., Brier). Even when not ranked first, such as on SVHN or CIFAR-100 on ViT with Transfer Learning, the performance gap to the top method is minimal, underscoring its consistent and robust performance. The use of Pareto plots, together with other metrics such as AdaptiveECE and Classwise-ECE in cases of similar error rate–ECE performance, provides a more suitable evaluation, as accuracy or calibration alone can be misleading: for instance, CCL-SC shows higher accuracy but worse calibration than Socrates for CIFAR-100 on VGG-16, whereas CCL-SC for SVHN on VGG-16 achieves lower accuracy despite better calibration. Beyond error rate and ECE, models trained with Socrates also demonstrate the best class-wise calibration across most dataset-architecture combinations. The models trained on SVHN highlight an interesting behaviour, as not all methods were able to adapt to this *toy dataset*, including CCL-SC, Focal, FLSD, and MC, and in some cases they did not reach competitive accuracy (CCL-SC and MC). By contrasting the results on SVHN with those on Food-101, we can argue that our method achieves strong performance in both simple and real-world scenarios, underscoring its flexibility and adaptability.

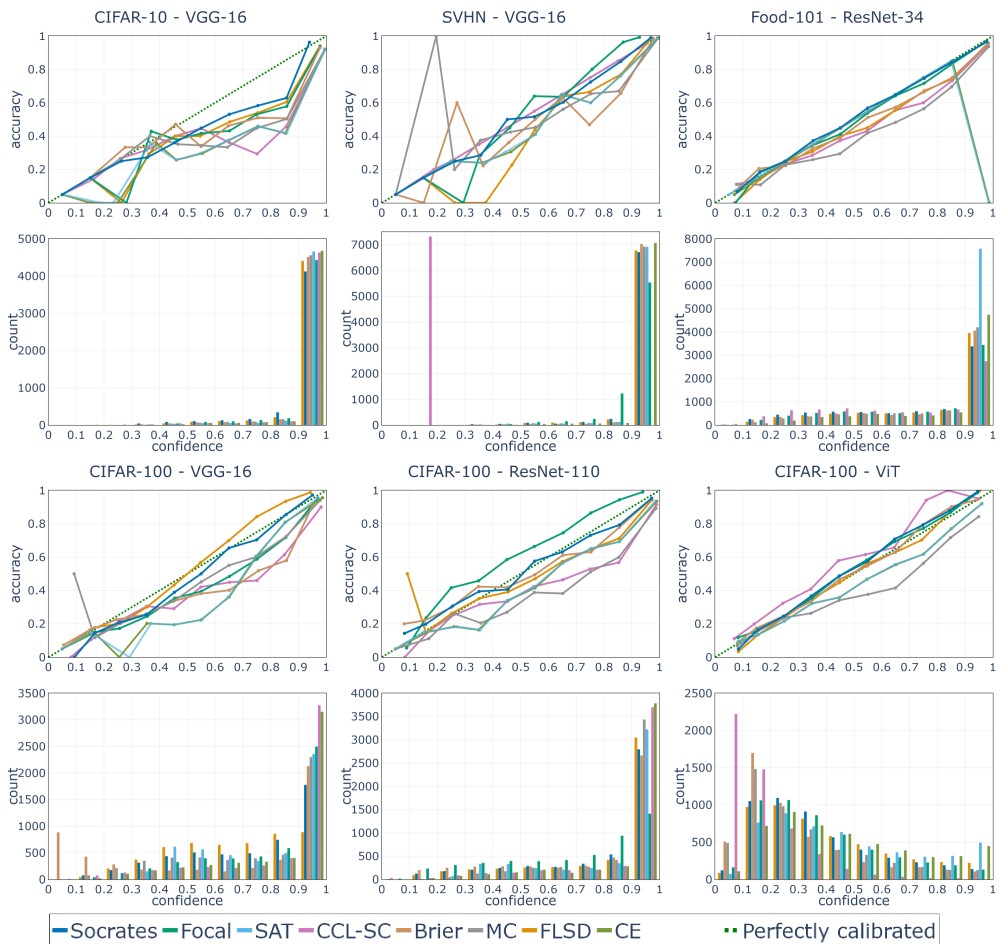

Figure 5: Reliability diagrams for the validation set at the final training epoch (epoch 300).

The Temperature Scaling (TS) and Matrix Scaling (MS) post-hoc methods alongside Socrates method (Table 2; Pareto plots in Figure 13 and Figure 14b in Appendix J) did not improve final calibration, underscoring their limitation: they were designed to correct overconfident predictions. In contrast, Vector Scaling (VS), with its per-class calibration, effectively reduces miscalibration in most-cases, demonstrating the need for post-hoc methods that handle heterogeneous confidence patterns. Conversely, CE can benefit from methods designed to correct overconfident predictions, such as Temperature Scaling. When comparing Socrates and CE with their respective post-hoc version, Socrates continues to achieve the best performance. The Food-101 and CIFAR-100 on ResNet-110 models further show that, in these settings, Socrates alone is a reliable confidence calibrator and classifier, without benefiting from additional post-hoc methods.

## 4.3 Ablation and Sensitivity Analysis

To validate the design of Socrates Loss and assess its practicality, we conducted an ablation study and analyzed its sensitivity to hyperparameters. Full results in Appendix G.

**Each Component of Socrates Loss is Essential.** Our primary design goal was to create a unified loss in which each component plays a critical role. To verify this, we carry out an ablation study systematically removing each key element of the loss function. The results confirm that removing any single component, i.e., the focal term, the adaptive target, or the dynamic uncertainty penalty ($\beta$), degrades mainly confidence

Table 1: Test set performance at epoch 300 for standard training and at epoch 50 for transfer learning (TL). Metrics reported: accuracy (acc), ECE, AdaptiveECE (AdaECE), and Classwise-ECE (CW-ECE). Poor performance using SAT on Food-101, and CCL-SC and MC on SVHN is attributed to premature convergence. Best results are highlighted in **bold**, and second-best are underlined.

| Dataset | | Metric | Socrates | SAT | CCL-SC | Focal | FLSD | Brier | MC |
|---|---|---|---|---|---|---|---|---|---|
| CIFAR-10 | VGG-16 | Acc | **88.42 ± 0.05** | 88.29 ± 0.34 | 87.92 ± 0.26 | 88.31 ± 0.19 | 88.16 ± 0.17 | 87.67 ± 0.10 | 87.53 ± 0.12 |
| | | ECE | **4.39 ± 0.25** | 9.00 ± 0.23 | 9.57 ± 0.36 | 6.27 ± 0.16 | 6.29 ± 0.16 | 7.21 ± 0.12 | 9.34 ± 0.17 |
| | | AdaECE | **6.03 ± 0.18** | 9.03 ± 0.23 | 9.50 ± 0.36 | 6.32 ± 0.25 | 6.38 ± 0.20 | 7.23 ± 0.15 | 9.34 ± 0.17 |
| | | CW-ECE | **1.31 ± 0.01** | 1.34 ± 0.05 | 1.55 ± 0.02 | 1.41 ± 0.03 | 1.42 ± 0.02 | 1.50 ± 0.03 | 1.49 ± 0.03 |
| SVHN | VGG-16 | Acc | 97.25 ± 0.08 | 97.21 ± 0.03 | 19.59 ± 0.00 | 66.08 ± 42.44 | 81.62 ± 34.68 | **97.31 ± 0.05** | 35.06 ± 34.59 |
| | | ECE | 2.49 ± 0.03 | 0.60 ± 0.04 | **0.57 ± 0.28** | 4.77 ± 3.15 | 2.31 ± 0.52 | 1.32 ± 0.12 | 2.70 ± 0.84 |
| | | AdaECE | 2.39 ± 0.06 | 0.74 ± 0.03 | **0.57 ± 0.28** | 4.72 ± 3.10 | 2.16 ± 0.59 | 2.18 ± 0.15 | 2.70 ± 0.84 |
| | | CW-ECE | **1.06 ± 0.00** | 1.07 ± 0.01 | 6.47 ± 0.49 | 3.20 ± 2.61 | 2.08 ± 2.23 | 1.07 ± 0.00 | 5.06 ± 2.23 |
| Food-101 | ResNet-34 | Acc | 77.72 ± 0.61 | 13.54 ± 30.28 | 73.61 ± 4.64 | 72.20 ± 12.10 | 74.91 ± 6.59 | **78.31 ± 0.40** | 75.39 ± 0.53 |
| | | ECE | **0.81 ± 0.21** | 80.98 ± 39.32 | 6.61 ± 0.70 | 0.83 ± 0.37 | 3.27 ± 0.09 | 3.08 ± 0.11 | 6.75 ± 0.37 |
| | | AdaECE | **0.82 ± 0.17** | 80.98 ± 39.32 | 6.60 ± 0.66 | 0.86 ± 0.31 | 3.22 ± 0.10 | 2.99 ± 0.16 | 6.74 ± 0.37 |
| | | CW-ECE | **0.23 ± 0.01** | 1.59 ± 0.60 | 0.27 ± 0.06 | 0.30 ± 0.15 | 0.26 ± 0.08 | 0.23 ± 0.01 | 0.24 ± 0.00 |
| CIFAR-100 | VGG-16 | Acc | 71.26 ± 0.21 | 66.14 ± 0.33 | **72.41 ± 0.37** | 71.93 ± 0.10 | 70.15 ± 0.32 | 53.87 ± 1.59 | 68.08 ± 0.17 |
| | | ECE | **3.45 ± 0.25** | 12.23 ± 0.31 | 11.91 ± 0.29 | 7.48 ± 0.37 | 5.36 ± 0.30 | 6.47 ± 0.57 | 8.00 ± 0.34 |
| | | AdaECE | **3.51 ± 0.23** | 12.40 ± 0.31 | 11.87 ± 0.31 | 7.49 ± 0.36 | 5.33 ± 0.26 | 7.06 ± 0.69 | 8.00 ± 0.34 |
| | | CW-ECE | **0.28 ± 0.00** | 0.49 ± 0.01 | 0.32 ± 0.01 | 0.28 ± 0.00 | 0.30 ± 0.00 | 0.51 ± 0.02 | 0.30 ± 0.01 |
| CIFAR-100 | ResNet-110 | Acc | 77.39 ± 0.32 | 75.41 ± 1.04 | **77.85 ± 0.67** | 76.62 ± 0.11 | 77.05 ± 0.42 | 74.31 ± 0.92 | 74.55 ± 0.88 |
| | | ECE | **2.86 ± 0.19** | 8.60 ± 0.28 | 10.40 ± 0.52 | 6.49 ± 1.72 | 4.15 ± 1.12 | 4.33 ± 0.23 | 13.51 ± 0.42 |
| | | AdaECE | **2.76 ± 0.28** | 8.60 ± 0.29 | 10.40 ± 0.52 | 6.46 ± 1.76 | 3.94 ± 1.08 | 4.18 ± 0.31 | 13.51 ± 0.42 |
| | | CW-ECE | **0.27 ± 0.01** | 0.33 ± 0.01 | 0.27 ± 0.01 | 0.31 ± 0.01 | 0.27 ± 0.01 | 0.30 ± 0.01 | 0.30 ± 0.01 |
| CIFAR-100 | ViT | Acc | 49.07 ± 3.20 | 42.82 ± 0.82 | 47.63 ± 5.16 | 48.30 ± 5.23 | **49.11 ± 4.21** | 42.85 ± 0.74 | 45.09 ± 9.04 |
| | | ECE | 4.10 ± 0.97 | 21.91 ± 0.73 | 11.66 ± 5.79 | 8.87 ± 3.38 | 10.58 ± 2.51 | **2.09 ± 0.32** | 6.65 ± 2.28 |
| | | AdaECE | 4.08 ± 1.04 | 21.91 ± 0.73 | 11.64 ± 5.82 | 8.86 ± 3.38 | 10.58 ± 2.51 | **1.96 ± 0.08** | 6.63 ± 2.29 |
| | | CW-ECE | **0.45 ± 0.03** | 0.73 ± 0.02 | 0.53 ± 0.11 | 0.50 ± 0.10 | 0.49 ± 0.07 | 0.54 ± 0.01 | 0.54 ± 0.14 |
| CIFAR-100 | ViT TL | Acc | 91.83 ± 0.06 | 71.27 ± 19.01 | **92.46 ± 0.09** | 69.05 ± 20.74 | 77.15 ± 18.65 | 58.55 ± 21.99 | 62.48 ± 26.89 |
| | | ECE | 5.30 ± 0.10 | 1.46 ± 0.06 | **0.64 ± 0.09** | 6.55 ± 1.23 | 1.68 ± 0.37 | 7.08 ± 3.28 | 1.60 ± 0.98 |
| | | AdaECE | 5.30 ± 0.10 | 1.44 ± 0.08 | **0.47 ± 0.15** | 6.60 ± 1.16 | 1.63 ± 0.35 | 7.10 ± 3.26 | 1.66 ± 0.84 |
| | | CW-ECE | 0.17 ± 0.00 | 0.34 ± 0.17 | **0.15 ± 0.00** | 0.39 ± 0.18 | 0.29 ± 0.18 | 0.52 ± 0.19 | 0.43 ± 0.26 |

calibration performance. In particular, the absence of the focal term from the ground truth component produced the worst outcomes, resulting in higher ECE. This effect is exacerbated when combined with the absence of either the adaptive target or $\beta$, further worsening classification and calibration performance. Moreover, testing alternative version of $\beta$ reinforces the value of the current $\beta$ approach. This finding underscores the importance of explicitly penalizing a model's failure to recognize its own uncertainty, and suggest that $\beta$ could potentially be studied as a standalone confidence calibration component in future work. Similarly, removing the adaptive target degrades performance, reinforcing the necessity of each element for achieving accurate and well-calibrated result.

**Low hyperparameter sensitivity.** Beyond its internal components, a practical loss function should ideally be robust to hyperparameter variations; however, some sensitivity is expected in confidence calibration tasks, where small changes can meaningfully affect predicted confidence. Our analysis reveals that Socrates Loss is robust to variations in its modularity factor $\gamma$ in terms of classification accuracy, though careful hyperparameter tuning can further improve calibration. In contrast, variations in the momentum factor $\alpha$ can impact performance, with higher values generally preferred. These hyperparameters are dataset-architecture dependent, and their optimal values can vary. The low sensitivity behavior contrasts sharply with the high sensitivity reported for the other methods, including the ones that combine losses in a two-phase training.

Table 2: Test set performance with post-hoc confidence calibration methods at epoch 300 for standard training and at epoch 50 for transfer learning (TL). Metrics reported: accuracy (acc), ECE, AdaptiveECE (AdaECE), and Classwise-ECE (CW-ECE). Best results are highlighted in **bold**.

| | Metric | Socrates | Socrates+TS | Socrates+MS | Socrates+VS | CE | CE+TS | CE+MS | CE+VS |
|---|---|---|---|---|---|---|---|---|---|
| **CIFAR-10 VGG-16** | Acc | $88.42 \pm 0.05$ | $88.40 \pm 0.04$ | $88.46 \pm 0.14$ | $88.37 \pm 0.06$ | $88.46 \pm 0.28$ | $88.52 \pm 0.28$ | $88.45 \pm 0.29$ | $\mathbf{88.55 \pm 0.37}$ |
| | ECE | $4.39 \pm 0.25$ | $8.32 \pm 0.33$ | $\mathbf{2.60 \pm 0.48}$ | $3.21 \pm 0.44$ | $9.09 \pm 0.24$ | $4.67 \pm 0.20$ | $2.78 \pm 0.10$ | $2.63 \pm 0.31$ |
| | AdaECE | $6.03 \pm 0.18$ | $8.90 \pm 0.14$ | $\mathbf{3.25 \pm 0.48}$ | $3.82 \pm 0.35$ | $9.08 \pm 0.23$ | $5.31 \pm 0.23$ | $3.58 \pm 0.23$ | $3.53 \pm 0.07$ |
| | CW-ECE | $1.31 \pm 0.01$ | $2.09 \pm 0.02$ | $\mathbf{1.29 \pm 0.01}$ | $1.30 \pm 0.01$ | $1.42 \pm 0.02$ | $1.39 \pm 0.04$ | $1.41 \pm 0.04$ | $1.42 \pm 0.05$ |
| **SVHN VGG-16** | Acc | $97.25 \pm 0.08$ | $97.24 \pm 0.09$ | $\mathbf{97.30 \pm 0.05}$ | $97.27 \pm 0.09$ | $97.19 \pm 0.04$ | $97.20 \pm 0.04$ | $97.16 \pm 0.05$ | $97.19 \pm 0.05$ |
| | ECE | $2.49 \pm 0.03$ | $6.91 \pm 0.05$ | $\mathbf{0.56 \pm 0.03}$ | $0.63 \pm 0.08$ | $1.58 \pm 0.05$ | $1.57 \pm 0.07$ | $0.71 \pm 0.05$ | $0.76 \pm 0.05$ |
| | AdaECE | $2.39 \pm 0.06$ | $6.88 \pm 0.06$ | $1.01 \pm 0.11$ | $1.15 \pm 0.05$ | $1.58 \pm 0.05$ | $1.54 \pm 0.11$ | $\mathbf{0.96 \pm 0.10}$ | $0.96 \pm 0.14$ |
| | CW-ECE | $1.06 \pm 0.00$ | $1.21 \pm 0.01$ | $0.97 \pm 0.00$ | $\mathbf{0.96 \pm 0.00}$ | $1.05 \pm 0.01$ | $1.09 \pm 0.00$ | $1.07 \pm 0.01$ | $1.07 \pm 0.01$ |
| **Food-101 ResNet-34** | Acc | $77.72 \pm 0.61$ | $77.68 \pm 0.70$ | $63.92 \pm 0.94$ | $77.66 \pm 0.80$ | $78.34 \pm 0.31$ | $\mathbf{78.38 \pm 0.35}$ | $48.71 \pm 31.82$ | $78.21 \pm 0.35$ |
| | ECE | $\mathbf{0.81 \pm 0.21}$ | $7.34 \pm 0.28$ | $35.65 \pm 0.96$ | $1.72 \pm 0.22$ | $7.28 \pm 0.25$ | $2.26 \pm 0.20$ | $34.93 \pm 0.39$ | $2.08 \pm 0.19$ |
| | AdaECE | $\mathbf{0.82 \pm 0.17}$ | $7.34 \pm 0.28$ | $35.65 \pm 0.96$ | $1.83 \pm 0.27$ | $7.28 \pm 0.25$ | $2.27 \pm 0.24$ | $34.93 \pm 0.39$ | $2.09 \pm 0.17$ |
| | CW-ECE | $0.23 \pm 0.01$ | $0.28 \pm 0.00$ | $0.69 \pm 0.02$ | $0.23 \pm 0.01$ | $\mathbf{0.22 \pm 0.00}$ | $0.24 \pm 0.01$ | $0.68 \pm 0.01$ | $0.24 \pm 0.00$ |
| **CIFAR-100 VGG-16** | Acc | $71.26 \pm 0.21$ | $71.31 \pm 0.20$ | $29.84 \pm 33.30$ | $71.43 \pm 0.18$ | $72.06 \pm 0.11$ | $\mathbf{72.10 \pm 0.08}$ | $58.77 \pm 0.41$ | $72.08 \pm 0.18$ |
| | ECE | $3.45 \pm 0.25$ | $9.41 \pm 0.21$ | $40.49 \pm 0.55$ | $\mathbf{2.67 \pm 0.09}$ | $13.17 \pm 0.10$ | $3.03 \pm 0.20$ | $40.53 \pm 0.38$ | $3.96 \pm 0.18$ |
| | AdaECE | $3.51 \pm 0.23$ | $9.41 \pm 0.21$ | $40.49 \pm 0.55$ | $\mathbf{2.66 \pm 0.10}$ | $13.16 \pm 0.12$ | $3.31 \pm 0.17$ | $40.53 \pm 0.38$ | $3.91 \pm 0.15$ |
| | CW-ECE | $\mathbf{0.28 \pm 0.00}$ | $0.34 \pm 0.01$ | $0.80 \pm 0.01$ | $\mathbf{0.28 \pm 0.00}$ | $0.32 \pm 0.00$ | $0.29 \pm 0.01$ | $0.80 \pm 0.01$ | $0.29 \pm 0.01$ |
| **CIFAR-100 ResNet-110** | Acc | $77.39 \pm 0.32$ | $77.44 \pm 0.35$ | $68.13 \pm 0.49$ | $77.31 \pm 0.23$ | $\mathbf{77.79 \pm 0.50}$ | $77.59 \pm 0.29$ | $51.68 \pm 33.79$ | $77.47 \pm 0.19$ |
| | ECE | $\mathbf{2.86 \pm 0.19}$ | $7.19 \pm 0.34$ | $31.49 \pm 0.50$ | $4.44 \pm 0.16$ | $10.17 \pm 0.81$ | $2.90 \pm 0.25$ | $31.03 \pm 0.64$ | $4.82 \pm 0.27$ |
| | AdaECE | $\mathbf{2.76 \pm 0.28}$ | $7.19 \pm 0.34$ | $31.49 \pm 0.50$ | $4.40 \pm 0.11$ | $10.14 \pm 0.85$ | $2.87 \pm 0.28$ | $31.03 \pm 0.64$ | $4.75 \pm 0.27$ |
| | CW-ECE | $0.27 \pm 0.01$ | $0.32 \pm 0.01$ | $0.62 \pm 0.01$ | $\mathbf{0.26 \pm 0.00}$ | $0.27 \pm 0.01$ | $0.29 \pm 0.01$ | $0.61 \pm 0.01$ | $\mathbf{0.26 \pm 0.00}$ |
| **CIFAR-100 ViT** | Acc | $\mathbf{49.07 \pm 3.20}$ | $48.50 \pm 3.49$ | $25.49 \pm 16.42$ | $47.91 \pm 3.66$ | $49.00 \pm 4.76$ | $48.41 \pm 5.28$ | $33.37 \pm 1.65$ | $47.92 \pm 5.31$ |
| | ECE | $4.10 \pm 0.97$ | $10.12 \pm 1.94$ | $52.35 \pm 25.65$ | $2.43 \pm 1.13$ | $16.42 \pm 2.49$ | $1.83 \pm 0.68$ | $55.54 \pm 22.97$ | $\mathbf{1.74 \pm 1.23}$ |
| | AdaECE | $4.08 \pm 1.04$ | $10.10 \pm 1.95$ | $52.35 \pm 25.65$ | $2.46 \pm 1.13$ | $16.42 \pm 2.49$ | $\mathbf{1.77 \pm 0.62}$ | $55.54 \pm 22.97$ | $1.83 \pm 1.17$ |
| | CW-ECE | $\mathbf{0.45 \pm 0.03}$ | $0.52 \pm 0.04$ | $1.12 \pm 0.36$ | $0.47 \pm 0.04$ | $0.55 \pm 0.08$ | $0.47 \pm 0.04$ | $1.18 \pm 0.32$ | $0.49 \pm 0.05$ |
| **CIFAR-100 ViT TL** | Acc | $\mathbf{91.83 \pm 0.06}$ | $91.81 \pm 0.05$ | $89.26 \pm 0.15$ | $91.47 \pm 0.05$ | $70.98 \pm 19.18$ | $74.50 \pm 20.20$ | $45.23 \pm 51.07$ | $75.85 \pm 18.15$ |
| | ECE | $5.30 \pm 0.10$ | $11.66 \pm 0.10$ | $10.32 \pm 0.12$ | $1.77 \pm 0.06$ | $\mathbf{1.49 \pm 0.37}$ | $7.60 \pm 0.52$ | $10.06 \pm 0.14$ | $1.96 \pm 0.28$ |
| | AdaECE | $5.30 \pm 0.10$ | $11.66 \pm 0.10$ | $10.32 \pm 0.12$ | $1.70 \pm 0.05$ | $\mathbf{1.52 \pm 0.32}$ | $7.58 \pm 0.49$ | $10.06 \pm 0.14$ | $1.90 \pm 0.17$ |
| | CW-ECE | $0.17 \pm 0.00$ | $0.21 \pm 0.00$ | $0.20 \pm 0.01$ | $\mathbf{0.14 \pm 0.01}$ | $0.34 \pm 0.17$ | $0.34 \pm 0.19$ | $0.20 \pm 0.01$ | $0.26 \pm 0.13$ |

## 5 Conclusion

This research addresses confidence calibration as an essential component for improving the reliability of Deep Neural Networks. We demonstrate that existing methods often lead to training instability, convergence failures, or a suboptimal accuracy-calibration performance. We answered our research questions affirmatively with the introduction of **Socrates Loss**, a novel easy-to-implement loss function that integrates uncertainty awareness directly into the training process. Supported by both theoretical analysis and empirical evidence, experiments conducted across four benchmarks and multiple architectures, evaluated using several metrics, show that Socrates Loss performs on par with or better than existing confidence calibration methods. Moreover, it also demonstrates better performance in terms of both training stability and accuracy-calibration trade-off. By avoiding the pitfalls of scheduled loss switching, Socrates Loss offers an effective alternative for training accurate and well-calibrated single models.

This work highlights several possible research directions. The training instability observed with the ViT architecture indicates that the interaction between confidence calibration methods and attention mechanisms requires further investigation. Furthermore, while Socrates Loss performs well in-distribution calibration, its robustness to distribution shifts and out-of-distribution (OOD) samples remains an open question. Its strong class-wise calibration suggests potential robustness to distribution shifts and positions it as a promising candidate for open-set recognition, although this hypothesis requires comprehensive empirical validation. The dynamic uncertainty penalty emerges as a key component of Socrates Loss, and could be further explored as a standalone regularizer or adapted to other reliability-related tasks such as OOD detection. Finally, since

Socrates Loss explicitly leverages model uncertainty through an unknown class, it could be further studied in the context of selective classification.

## Acknowledgments

The authors wish to acknowledge use of the eResearch Infrastructure Platform hosted by the Crown company, Research and Education Advanced Network New Zealand (REANNZ) Ltd., and funded by the New Zealand Ministry of Business, Innovation and Employment (MBIE). This research was supported by use of the Nectar Research Cloud, accessed through the University of Auckland. The Nectar Research Cloud is a collaborative Australian research platform supported by the NCRIS-funded Australian Research Data Commons (ARDC). This research was funded by MBIE through the Science for Technological Innovation National Science Challenge, contract id 2021-SfTI-SpH10-UoA.

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

# A    Appendix Content

- Appendix B: Calibration Evaluation Methods
  - Appendix B.1: Visualization Metric
    * Appendix B.1.1: Reliability Diagrams
    * Appendix B.1.2: Pareto Plots
  - Appendix B.2: Quantitative Calibration Metrics
    * Appendix B.2.1: Expected Calibration Error (ECE)
    * Appendix B.2.2: Maximum Calibration Error (MCE)
    * Appendix B.2.3: AdaptiveECE (AdaECE)
    * Appendix B.2.4: Classwise-ECE (CW-ECE)

- Appendix C: Socrates Loss - Pseudocode

- Appendix D: Socrates Loss - Mathematical example

- Appendix E: Theoretical Proofs
  - Appendix E.1: Socrates Loss Regularizes the Weights of the Network
  - Appendix E.2: Socrates Loss Forms a Regularized Upper Bound on the Kullback-Leibler Divergence

- Appendix F: Model Reproducibility
  - Appendix F.1: Further Details on Model Training
  - Appendix F.2: Hyperparameter Tuning and Final Hyperparameters

- Appendix G: Hyperparameter Sensitivity Analysis, Alternative Dynamic Uncertainty Penalties, and Ablation Study
  - Appendix G.1: Hyperparameter Sensitivity
  - Appendix G.2: Exploring Alternative Dynamic Uncertainty Penalties
  - Appendix G.3: Ablation Study

- Appendix H: Unknown Class Behaviour During Training

- Appendix I: Validation Set Performance

- Appendix J: Test Set Performance - Pareto Plots

# B  Calibration Evaluation Methods

This section provides formal definitions for the calibration visualization and quantitative metrics used in the evaluation.

## B.1  Visualization Metrics

Reliability diagrams and Pareto plots are described and discussed in detail in this subsection. Examples of both are provided in Fig. 4 (main text).

### B.1.1  Reliability Diagrams

The level of confidence calibration can be assessed visually using a reliability diagram (Niculescu-Mizil & Caruana, 2005). This diagram plots the expected sample accuracy as a function of prediction confidence at a single epoch. To generate the diagram, confidences are grouped into $M$ interval bins. Following the methodology of Guo et al. (2017), we use fixed-width bins of size $1/M$. For each bin $B_m$, which contains the set of indices for samples whose confidence $\hat{p}_i$ falls into the interval $I_m = (\frac{m-1}{M}, \frac{m}{M}]$, we calculate the average bin confidence and average bin accuracy.

The average confidence for bin $B_m$ is defined as:

$$\text{conf}(B_m) = \frac{1}{|B_m|} \sum_{i \in B_m} \hat{p}_i \tag{6}$$

The average accuracy for bin $B_m$ is defined as:

$$\text{acc}(B_m) = \frac{1}{|B_m|} \sum_{i \in B_m} \mathbf{1}(\hat{y}_i = y_i) \tag{7}$$

where $\hat{p}_i$ is the confidence for the $i$-th instance, $\hat{y}_i$ is its predicted class label, and $y_i$ is its true class label for the $i$-th sample. For a perfectly calibrated model, the plot of $\text{acc}(B_m)$ vs. $\text{conf}(B_m)$ would form the identity function (perfectly calibrated line).

### B.1.2  Pareto Plots

While reliability diagrams are a standard tool for visualizing confidence calibration, they offer a limited view by collapsing confidence calibration and accuracy into a single curve and analyzing one unique epoch. In contrast, we propose the use of Pareto plots, which provide a richer and more informative perspective by explicitly illustrating the trade-off between the error rate (1-accuracy) and any calibration metric, such as the Expected Calibration Error (ECE), across training epochs, enabling more nuanced model comparisons. This is especially important in settings where improvements in one metric may come at the cost of the other. Furthermore, they align naturally with the multi-objective nature of real-world applications, where both confidence calibration and predictive performance are critical. By identifying the point closest to the origin (i.e., bottom-left), one can select models that simultaneously minimize both error and miscalibration.

## B.2  Quantitative Calibration Metrics

While reliability diagrams and Pareto plots offer visual insight, quantitative metrics are required for rigorous comparison. In this subsection, we examine the Expected Calibration Error (ECE), Maximum Calibration Error (MCE), AdaptiveECE (AdaECE), and Classwise-ECE (CW-ECE). Lower values of these metrics indicate better confidence calibration.

### B.2.1 Expected Calibration Error (ECE)

The Expected Calibration Error (ECE) (Naeini et al., 2015) quantifies the overall calibration error by computing the weighted average of the bin-wise difference between accuracy and confidence:

$$\text{ECE} = \sum_{m=1}^{M} \frac{|B_m|}{n} |\text{acc}(B_m) - \text{conf}(B_m)| \tag{8}$$

where $n$ is the total number of samples.

In other words, this corresponds to the average vertical gap between the reliability curve and the ideal identity line in a reliability diagram.

### B.2.2 Maximum Calibration Error (MCE)

For high-stakes applications, the worst-case deviation is often more critical than the average error. The Maximum Calibration Error (MCE) (Naeini et al., 2015) captures this by identifying the largest calibration deviation across all bins:

$$\text{MCE} = \max_{m \in \{1,...,M\}} |\text{acc}(B_m) - \text{conf}(B_m)| \tag{9}$$

Put differently, this corresponds to the biggest vertical gap between the reliability curve and the ideal identity line in a reliability diagram.

In this research, we do not consider the MCE, as it can be disproportionately influenced by sparsely populated bins or rare events, resulting in unstable or misleading assessments of model calibration. While MCE may provide complementary insights in high-stakes scenarios where outlier behavior is critical, it should be interpreted with caution and not relied upon in isolation.

### B.2.3 AdaptiveECE (AdaECE)

A known limitation with ECE is its susceptibility to bias from the fixed-width binning scheme, as bins with more samples have a greater influence and tend to dominate the final score (Mukhoti et al., 2020). To mitigate this, AdaptiveECE (AdaECE) (Mukhoti et al., 2020) adaptively modifies the binning strategy to ensure each bin contains an equal number of samples, thus providing a more balanced assessment of confidence calibration error. The formula remains the same as ECE, but the bins $B_m$ are constructed such that $|B_m|$ is constant for all $m$, i.e., $\forall m, m' \in M : |B_m| = |B_{m'}|$.. Nevertheless, this method still requires specifying the number of bins a priori.

### B.2.4 Classwise-ECE (CW-ECE)

ECE only considers the confidence of the top predicted class, then, calibration of more populated classes may overshadow classes with less instances, potentially hiding miscalibration in underrepresented classes. To provide a more comprehensive measure, Classwise-ECE (CW-ECE) (Mukhoti et al., 2020) extends the calculation to all classes, providing a more fine-grained evaluation. It computes the ECE for each class individually and averages the results:

$$\text{CW-ECE} = \frac{1}{K} \sum_{k=1}^{K} \left( \sum_{m=1}^{M} \frac{|B_{m,k}|}{n} |\text{acc}(B_{m,k}) - \text{conf}(B_{m,k})| \right) \tag{10}$$

where $K$ is the number of classes and $B_{m,k}$ represents the samples in bin $m$ for class $k$.

## C  Socrates Loss - Pseudocode

---

**Algorithm 1** Training with Socrates Loss

---

**Require:** Data $\{(x_i, y_i)\}_{i=1}^n$, architecture of the model $f$, mini-batch size $BM$, and Socrates hyperparameters: momentum factor $\alpha$, modularity factor $\gamma$, and initial epochs $E_s$.

1: **for** $e = 0$ **to** maximum_epochs$-1$ **do**
2:   **for** each mini-batch data $\{(x_i, y_i)\}_{BM}$ in the current epoch $e$ **do**
3:     **for** $i = 1$ **to** BM **(in parallel) do**
4:       **if** $e = 0$ **then**
5:         $t_{i,y_i,e} = y_i$
6:       **end if**
7:       $\hat{p}_i = softmax(f(x_i))$
8:       $\beta_{i,e} = \max_{\bar{y}_i \neq y_i} (\hat{p}_{i,\bar{y}_i,e}) - \hat{p}_{i,idk,e}$
9:       **if** $e \geq E_s$ **then**
10:        $t_{i,y_i,e} = \alpha \times t_{i,y_i,e-1} + (1 - \alpha) \times \hat{p}_{i,y_i,e}$
11:       **end if**
12:       $\mathcal{L}_{\text{Socrates}}(f) = -\frac{1}{n} \sum_{i=1}^n (1 - \hat{p}_{i,y_i,e})^\gamma [t_{i,y_i,e} \log \hat{p}_{i,y_i,e} + \beta_{i,e}(1 - t_{i,y_i,e}) \log \hat{p}_{i,idk,e}]$
13:       Update the weights of $f$ using an optimizer based on $\mathcal{L}_{\text{Socrates}}(f)$
14:     **end for**
15:   **end for**
16: **end for**

---

Although our method allows for a combination of losses due to the flexibility of the initial epochs variable, we set $E_s = 0$ to avoid the combination and potential stability issues, as observed in the baseline methods SAT and CCL-SC.

## D  Socrates Loss - Mathematical example

To illustrate how Socrates loss operates, consider a model with $E_s = 0$, $\gamma = 2$, and $\alpha = 0.9$, capable of classifying into predator (class 0), non-predator (class 1), or unknown (class 3). We analyze three scenarios:

1. An image of a cat with a ground truth (gt) label of predator. The loss at epoch 31 is $\Rightarrow$ At epoch 30, the classifier outputs $[0.9, 0.05, 0.05]$ confidences, resulting in $t_{i,y_i,e-1} = 0.9$ based on previous predictions. At epoch 31, the classifier outputs $[0.9, 0.02, 0.08]$, updating $t_{i,y_i,e} = 0.9 \times 0.9 + (1 - 0.9) \times 0.9 = 0.9$, which remains high due to the high confidence at epoch 30. Then, since $\max_{\bar{y}_i \neq y_{gt}} \hat{p}_{i,\bar{y}_i,e}$ is the unknown class, $\beta = 0.08 - 0.08 = 0$. Finally, the loss at epoch 31 is $\mathcal{L}_{\text{Soc}}(f) = (1 - 0.9)^2 [0.9 \log 0.9 + 0 \times (1 - 0.9) \log 0.9] = -(1 - 0.9)^2 \times 0.9 \log 0.9 = 0.0009$. Thus, only the ground truth part and the focal term are relevant, penalizing hard-to-classify instances more.

2. An image of a pink cat with a gt label of predator. The loss at epoch 31 is $\Rightarrow$ At epoch 30, the classifier outputs $[0.5, 0.25, 0.25]$ with $t_{i,y_i,e-1} = 0.5$. At epoch 31, the classifier outputs $[0.5, 0.3, 0.2]$ and $t_{i,y_i,e} = 0.9 \times 0.5 + (1 - 0.9) \times 0.5 = 0.5$, which is not high as previous prediction lacked high confidence. Therefore, both parts in the loss equation are relevant. Since $\max_{\bar{y}_i \neq y_{gt}} \hat{p}_{i,\bar{y}_i}$ is the non-predator class, then $\beta = 0.3 - 0.2 = 0.1$; the model is unaware of its lack of knowledge. Thus, the loss at epoch 31 is $\mathcal{L}_{\text{Soc}}(f) = (1 - 0.5)^2 [0.5 \log 0.5 + 0.1 \times (1 - 0.5) \log 0.2] = 0.11$.

3. An image of a pink cat toy with a gt label of predator. The loss at epoch 31 is$\Rightarrow$ At epoch 30, the classifier outputs $[0.5, 0.25, 0.25]$, and a $t_{i,y_i,e-1} = 0.5$. At epoch 31 the model outputs $[0.5, 0.2, 0.3]$ and $t_{i,y_i,e} = 0.9 \times 0.5 + (1 - 0.9) \times 0.5 = 0.5$. Then, as previous predictions lacked high confidence, both parts of the equation take relevance. Since $\max_{\bar{y}_i \neq y_{gt}} \hat{p}_{i,\bar{y}_i}$ is the unknown class, then $\beta = 0.3 - 0.3 = 0$; the model is aware of its lack of knowledge. Finally, at epoch 31 is $\mathcal{L}_{\text{Soc}}(f) = (1 - 0.5)^2 [0.5 \log 0.5 + 0 \times (1 - 0.5) \log 0.2] = 0.087$.

These three scenarios illustrate the main functioning of the Socrates loss: the loss is smaller when current and previous predictions are close to the gt class (scenario 1) and higher when predictions deviate from it (scenarios 2 and 3). Additionally, when the classifier is uncertain about its own lack of knowledge, the loss increases, penalizing the classifier (scenario 2).

The decision to exclude the gt class when computing $\beta$ in Socrates loss is rooted in our goal to penalize cases where the model lacks certainty. Including the gt class as the maximum could penalize the model without reflecting its actual ability to recognize uncertainty. For instance, in scenarios 2 and 3, including the gt would result in a penalty due to the dynamic uncertainty term in both cases. However, we aim to penalize the model when its awareness of uncertainty is low or decreases, i.e., when the probability of the unknown class is not the highest, indicating the model does not recognize its own uncertainty despite some knowledge of the gt class.

## E    Theoretical Proofs

In this section, we provide the proofs for the theoretical claims presented in the main text.

### E.1    Socrates Loss Regularizes the Weights of the Network

**Theorem:**    Let $\mathcal{L}_{\mathrm{CE}}(f)$ be cross-entropy loss (CE), and $\mathcal{L}_{\mathrm{Soc}}(f)$ be Socrates loss. The gradients of the neural network trained with $\mathcal{L}_{\mathrm{Soc}}(f)$ are smaller than the ones trained with $\mathcal{L}_{\mathrm{CE}}(f)$ when smaller confidence is reached and the model could start overfitting and subsequently be miscalibrated, i.e.,

$$||\frac{\partial \mathcal{L}_{\mathrm{Soc}}(f)}{\partial w}|| \leq ||\frac{\partial \mathcal{L}_{\mathrm{CE}}(f)}{\partial w}||. \tag{11}$$

This behaviour shows that Socrates loss acts as a regularizer when the model is sufficiently confident, avoiding miscalibration and overfitting.

**Proof:**    To simplify, we consider the case of the first selected epochs where $t_i \leftarrow y_i = 1$. If we take only one instance from $m$ instances, i.e., $m = 1$, Socrates loss is:

$$\mathcal{L}_{\mathrm{Soc}}(f) = -\left[ t_y(1 - \hat{p}_y)^\gamma \log \hat{p}_y + \beta(1 - t_y)(1 - \hat{p}_y)^\gamma \log \hat{p}_{idk} \right]. \tag{12}$$

The gradient with respect to the parameters of the last linear layer can be decomposed with the chain rule:

$$\frac{\partial \mathcal{L}_{\mathrm{Soc}}(f)}{\partial w} = \frac{\partial \mathcal{L}(f)}{\partial \hat{p}_y} \frac{\partial \hat{p}_y}{\partial z} \frac{\partial z}{\partial w};$$

$$\text{where } \frac{\partial \mathcal{L}_{\mathrm{Soc}}(f)}{\partial \hat{p}_y} = \gamma(1 - \hat{p}_y)^{\gamma-1} t_y \log \hat{p}_y - (1 - \hat{p}_y)^\gamma \frac{t_y}{\hat{p}_y} + \tag{13}$$

$$+\gamma(1 - \hat{p}_y)^{\gamma-1}\beta(1 - t_y) \log \hat{p}_{idk} - (1 - \hat{p}_y)^\gamma \beta(1 - t_y) \frac{1}{\hat{p}_{idk}}.$$

On the other hand, CE loss is $\mathcal{L}_{\mathrm{CE}}(f) = -t_y \log \hat{p}_y$. Where the gradient using the chain rule is:

$$\frac{\partial \mathcal{L}_{\mathrm{CE}}(f)}{\partial w} = \frac{\partial \mathcal{L}_{\mathrm{CE}}(f)}{\partial \hat{p}_y} \frac{\partial \hat{p}_y}{\partial z} \frac{\partial z}{\partial w}; \text{ where } \frac{\partial \mathcal{L}_{\mathrm{CE}}(f)}{\partial \hat{p}_y} = -\frac{t_y}{\hat{p}_y} \tag{14}$$

We observe that the gradient of CE is a component of the gradient of Socrates:

$$\frac{\partial \mathcal{L}_{\mathrm{Soc}}(f)}{\partial \hat{p}_y} = \frac{\partial \mathcal{L}_{\mathrm{CE}}(f)}{\partial \hat{p}_y}[(1 - \hat{p}_y)^\gamma - \gamma\hat{p}_y(1 - \hat{p}_y)^{\gamma-1} \log \hat{p}_y]+$$

$$+\gamma(1 - \hat{p}_y)^{\gamma-1}\beta(1 - t_y) \log \hat{p}_{idk} - (1 - \hat{p}_y)^\gamma \beta(1 - t_y)\frac{1}{\hat{p}_{idk}}. \tag{15}$$

If $g(\hat{p}_y, \gamma) = (1 - \hat{p}_y)^\gamma - \gamma \hat{p}_y (1 - \hat{p}_y)^{\gamma-1} \log \hat{p}_y$ is a regularizer of the CE; and $r(t_y, \beta, \hat{p}_y, \hat{p}_{idk}) = = \gamma (1 - \hat{p}_y)^{\gamma-1} \beta (1 - t_y) \log \hat{p}_{idk} - (1 - \hat{p}_y)^\gamma \beta (1 - t_y) \frac{1}{\hat{p}_{idk}}$ is highly affected by the idk class, which adds a small penalty $r(t_y, \beta, \hat{p}_y, \hat{p}_{idk}) \in [0, 1]$, then:

$$\frac{\partial \mathcal{L}_{\text{Soc}}(f)}{\partial \hat{p}_y} = \frac{\partial \mathcal{L}_{\text{CE}}(f)}{\partial \hat{p}_y} g(\hat{p}_y, \gamma) + r(t_y, \beta, \hat{p}_y, \hat{p}_{idk}). \tag{16}$$

When confidence is high, and the model could start being overfitted and miscalibrated, the value of $g(\hat{p}_y, \gamma) \in [0, 1]$. In that case:

$$||\frac{\partial \mathcal{L}_{\text{Soc}}(f)}{\partial \hat{p}_y}|| \leq ||\frac{\partial \mathcal{L}_{\text{CE}}(f)}{\partial \hat{p}_y}|| \implies ||\frac{\partial \mathcal{L}_{\text{Soc}}(f)}{\partial w}|| \leq ||\frac{\partial \mathcal{L}_{\text{CE}}(f)}{\partial w}|| \tag{17}$$

This demonstrates that the gradients of a model associated with the Socrates loss are smaller than those associated with the CE when perfect confidence is reached. Therefore, the Socrates loss acts as a regularizer with a penalty associated with the unknown knowledge of the classifier, avoiding overfitting, and subsequently miscalibration.

### E.2 Socrates Loss Forms a Regularized Upper Bound on the Kullback-Leibler Divergence

**Theorem:** Socrates loss minimizes (creates an upper bound for) the Kullback-Leibler (KL) divergence while regularizing by increasing the entropy of the predicted distribution and leveraging the predictions associated with the unknown class. The regularization parameters are $\gamma, \beta$, and $\triangle_{reg}$; where $\triangle_{reg} = (1 - t_y)[\gamma \hat{p}_y \log \hat{p}_{idk} - \log \hat{p}_{idk}]$. Therefore: $\mathcal{L}(f) \geq KL(q\|\hat{p}) - \gamma \mathbb{H}[\hat{p}] + \beta \triangle_{reg}$;

**Proof:** Let the KL divergence be the divergence between the ground truth distribution $q$ and the predicted distribution $\hat{p}$, and $\mathbb{H}[q]$ be the entropy of the ground truth distribution defined as $\mathbb{H}[q] = -\sum_j q_j \log(q_j)$. Therefore, for a multiclass problem, the KL divergence can be expressed as:

$$KL(q\|\hat{p}) = \sum_j q_j \log(\frac{q_j}{\hat{p}_j}) = \sum_j q_j \log(q_j) - \sum_j q_j \log(\hat{p}_j); \Rightarrow KL(q\|\hat{p}) = -\mathbb{H}[q] + \mathcal{L}_{\text{CE}}(f); \tag{18}$$

where $\mathcal{L}_{\text{CE}}(f)$ is the cross-entropy loss (CE), which forms an upper bond on the KL divergence: $\mathcal{L}_{\text{CE}}(f) = KL(q\|\hat{p}) + \mathbb{H}[q]; \Rightarrow \mathcal{L}_{\text{CE}}(f) \geq KL(q\|\hat{p})$.

To simplify, we consider the case of the first selected epochs where $t_i \leftarrow y_i = 1$. Let $t_i \in q$, be the target distribution. If we take only one instance of m number of instances, i.e., $m = 1$, the loss function can be written as:

$$\mathcal{L}_{\text{Soc}}(f) = - [t_y (1 - \hat{p}_y)^\gamma \log \hat{p}_y + \beta (1 - t_y)(1 - \hat{p}_y)^\gamma \log \hat{p}_{idk}], \tag{19}$$

where the subscript $y$ denotes the values associated with the ground truth class and $idk$ the values associated with the extra unknown class.

Using Bernoulli's inequality, which states that $(1 - x)^\alpha \geq 1 - \alpha x$, if $0 \leq x \leq 1$ and $\alpha \geq 0$, as $\forall \gamma \geq 1$ and the $\hat{p}_y \in [0, 1]$, then we get:

$$\begin{aligned}
\mathcal{L}_{\text{Soc}}(f) = -(1 - \hat{p}_y)^\gamma [t_y \log \hat{p}_y + \beta (1 - t_y) \log \hat{p}_{idk}] &\geq -(1 - \gamma \hat{p}_y)[t_y \log \hat{p}_y + \beta (1 - t_y) \log \hat{p}_{idk}] = \\
&= \gamma \hat{p}_y t_y \log \hat{p}_y - t_y \log \hat{p}_y + \gamma \hat{p}_y \beta (1 - t_y) \log \hat{p}_{idk} - \beta (1 - t_y) \log \hat{p}_{idk} = \\
&= -\gamma \mathbb{H}[\hat{p}] + \mathcal{L}_{\text{CE}}(f) + \beta \triangle_{reg} = -\gamma \mathbb{H}[\hat{p}] + KL(q\|\hat{p}) + \mathbb{H}[q] + \beta \triangle_{reg}; \\
&\text{where } \triangle_{reg} = (1 - t_y)[\gamma \hat{p}_y \log \hat{p}_{idk} - \log \hat{p}_{idk}];
\end{aligned} \tag{20}$$

$\triangle_{reg}$ is considered a regularization term, as it is derived from a different distribution, the idk distribution, rather than the ground truth distribution. Using the Bernouilli inequality its error terms are typically small, especially in higher-order deviations. However, as the problem complexity increases, these errors can accumulate and become significant, particularly in high-dimensional spaces (curse of dimensionality). In this

proof, we have neglected these errors, and we have not provided a detailed error analysis. We acknowledge that these accumulated errors may affect the model's stability and convergence.

Therefore:

$$\mathcal{L}_{\mathrm{Soc}}(f) \geq KL(q\|\hat{p}) + \mathbb{H}[q] - \gamma\mathbb{H}[\hat{p}] + \beta\triangle_{reg}; \tag{21}$$

where $\mathbb{H}[q]$ is a constant.

Thus, this new loss improves confidence calibration by minimizing the KL divergence, maximizing the entropy depending on the weight of $\gamma$ (which smooths the learned distributions), and adding an extra regularization term (which might help to avoid overfitting) which maximises the uncertainty when the prediction is incorrect.

## F   Model Reproducibility

This section provides further details on model training, and additionally describes the hyperparameter tuning process and the final selected hyperparameters.

### F.1   Further Details on Model Training

The experiments were conducted on a shared supercomputer (Nvidia A100 80Gb SXM4 GPU). Note we do not provide run times for each method due to the nature of a shared supercomputer, where training durations vary based on resource availability.

The models were trained using Stochastic Gradient Descent (SGT) with an initial learning rate of 0.1 and a momentum of 0.9. The learning rate was reduced by 0.5 every 25 epochs. Weight decay was set to 0.0005. For the transfer learning setting, we adopt a two-group optimization strategy. The ViT backbone is fine-tuned with a learning rate of 0.0005 scaled by a factor of 0.1, ensuring slower and more stable updates to the pre-trained weights. In contrast, the newly initialized classification head is trained with a 0.0005 learning rate, allowing for faster adaptation to the target dataset. The learning rate was reduced by 0.5 at epochs 20, 35, and 45. Both parameter groups are optimized using Stochastic Gradient Descent with a 0.9 momentum and a 0.0005 weight decay.

For SAT and Socrates methods, an additional class, the unknown class, was included.

The same data augmentation techniques were applied uniformly across all methods for each dataset. We utilized widely adopted data augmentation strategies specifically designed for these datasets from the image classification domain. For the training sets, we used RandomCrop, RandomHorizontalFlip, and Normalize for CIFAR-10 and CIFAR-100; RandomRotation, RandomCrop, and Normalize for SVHN; and RandomResizedCrop, RandomHorizontalFlip, and Normalize for Food-101. For the validation and test sets, we applied Normalize for CIFAR-10, CIFAR-100, and SVHN; and CenterCrop followed by Normalize for Food-101.

The CCL-SC code was modified to ensure correct functionality, specifically by initializing the variables *temp_full_k1* and *temp_full_k2* as *False*.

Further implementation details are available in the code repository.

### F.2   Hyperparameter Tuning and Final Hyperparameters

To tune the hyperparameters, we used the full training and validation sets for each dataset with five seeds, except for Food-101, for which only three seeds were used due to its high computational cost.

For the Socrates method, we tested the modularity factor $\gamma \in \{1, 2, 3, 4\}$, momentum factor $\alpha \in \{0.8, 0.9, 0.99, 0.999\}$, and mini-batch train $MB \in \{64, 128\}$, with initial epochs $E_s = 0$.

For the other methods, we followed their original settings; when unspecified, we applied our hyperparameter search strategy on the ranges provided by the authors. For CCL-SC, we tuned the momentum coefficient $q \in \{0.9, 0.99, 0.999\}$, weight coefficient $w \in \{0.1, 0.5, 1.0\}$, queue size $s \in \{300, 3000, 10000\}$, initial epochs $E_s \in \{25, 50, 100, 150, 200\}$, and mini-batch train $MB \in \{64, 128\}$, while the MOCO dimension (Mdim) was varied depending on the dataset. For SAT, the momentum term $m \in \{0.9, 0.99, 0.999\}$, initial epochs

$E_s \in \{0, 25, 50, 100, 150, 200\}$, and mini-batch train $MB \in \{64, 128\}$. For Focal and FLSD, the modularity factor $\gamma \in \{1, 2, 3\}$ and mini-batch train $MB \in \{64, 128\}$ with initial epochs $E_s = 0$. For Brier Score and MC a mini-batch train $MB \in \{64, 128\}$ with initial epochs $E_s = 0$.

The final selection of hyperparameters is provided in Table 3.

An example of the behaviour of all tested hyperparameters for CIFAR-100 with VGG-16 (averaged over five seeds) is shown in Fig. 6. The Figure illustrates the impact of each hyperparameter on model performance. In this case, higher modularity factors and mid-range momentum factors help to train confidence calibrated models with competitive accuracy. The rationale behind this behavior is discussed in Subsection 3.2 (main text). However, as shown in Table 3, the optimal hyperparameter selection is dataset and architecture dependent. A complete hyperparameter sensitivity analysis is provided in Appendix G.

Table 3: Final hyperparameters. Underlined values are from their original research.

| Method | Hyperp. | CIFAR-10 VGG-16 | SVHN VGG-16 | Food-101 ResNet-34 | CIFAR-100 VGG-16 | CIFAR-100 ViT | CIFAR-100 ViT TL | CIFAR-100 ResNet-110 |
|---|---|---|---|---|---|---|---|---|
| Socrates | $\gamma$ | 2 | 1 | 2 | 2 | 4 | 1 | 1 |
| | $\alpha$ | 0.8 | 0.9 | 0.999 | 0.999 | 1 | 0.999 | 0.999 |
| | $E_s$ | 0 | 0 | 0 | 0 | 0 | 0 | 0 |
| | MB | 128 | 128 | 128 | 128 | 128 | 128 | 128 |
| CCL-SC | $q$ | 0.999 | 0.99 | 0.9 | 0.99 | 0.9 | 0.9 | 0.9 |
| | $w$ | 0.5 | 1.0 | 0.1 | 1.0 | 0.1 | 0.1 | 0.1 |
| | $s$ | 300 | 3000 | 3000 | 3000 | 3000 | 3000 | 3000 |
| | Mdim | 512 | 512 | 4096 | 512 | 512 | 512 | 2048 |
| | $E_s$ | 150 | 150 | 150 | 150 | 150 | 25 | 150 |
| | MB | 64 | 64 | 64 | 64 | 64 | 64 | 64 |
| SAT | $m$ | 0.90 | 0.90 | 0.90 | 0.90 | 0.99 | 0.90 | 0.99 |
| | $E_s$ | 0 | 100 | 0 | 200 | 200 | 25 | 150 |
| | MB | 128 | 128 | 128 | 128 | 128 | 128 | 128 |
| Focal | $\gamma$ | 1 | 2 | 3 | 1 | 2 | 2 | 3 |
| | $E_s$ | 0 | 0 | 0 | 0 | 0 | 0 | 0 |
| | MB | 128 | 128 | 128 | 128 | 128 | 128 | 128 |
| FLSD | $\gamma$ | 1 | 1 | 1 | 3 | 1 | 2 | 1 |
| | $E_s$ | 0 | 0 | 0 | 0 | 0 | 0 | 0 |
| | MB | 128 | 128 | 128 | 128 | 128 | 128 | 128 |
| MC & Brier Score | $E_s$ | 0 | 0 | 0 | 0 | 0 | 0 | 0 |
| | MB | 128 | 128 | 128 | 128 | 128 | 128 | 128 |

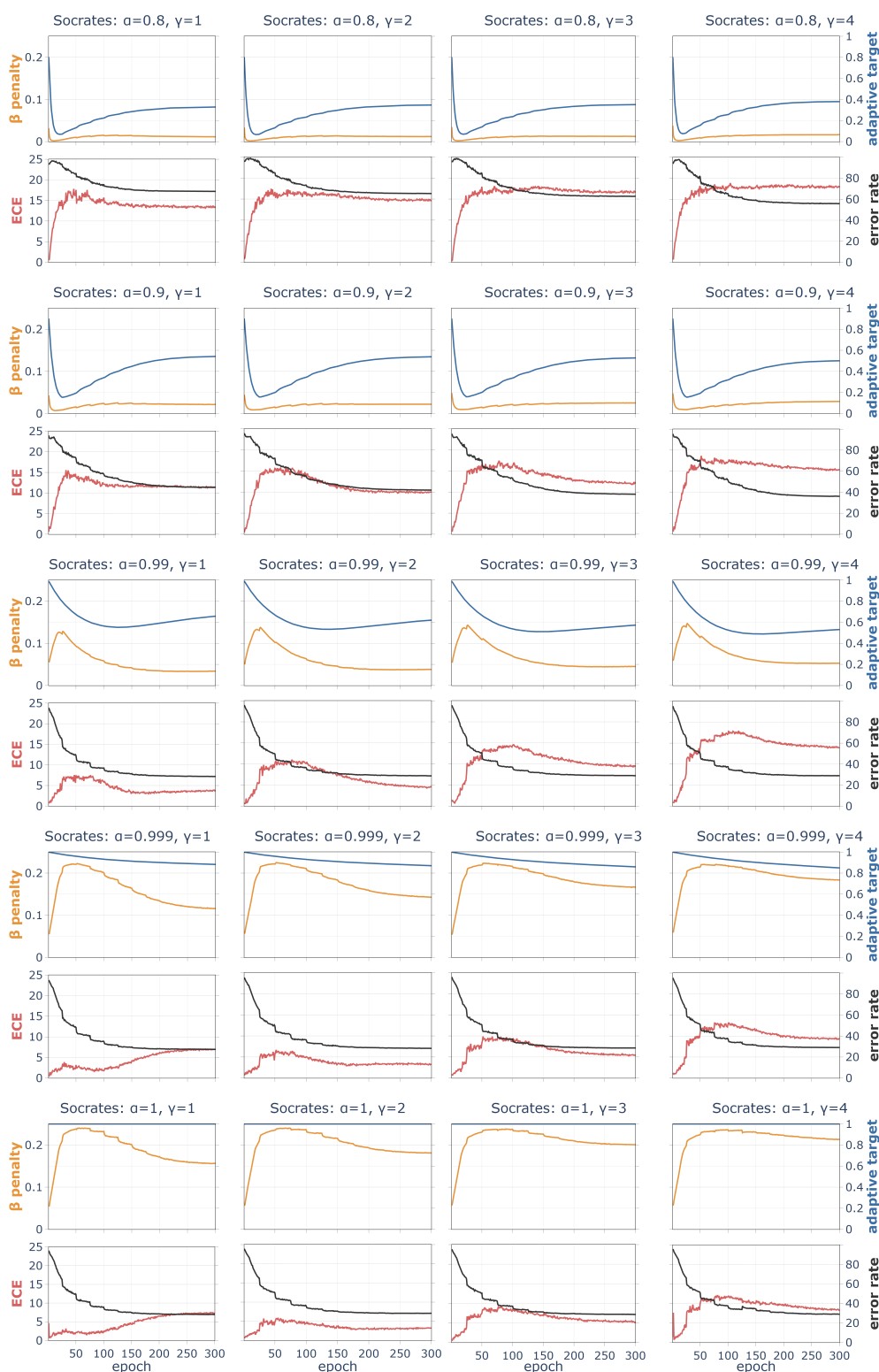

Figure 6: Evolution of the Socrates method on CIFAR-100 with VGG-16 for all hyperparameter configurations. The curves show the mean values across epochs of the dynamic uncertainty penalty ($\beta$ penalty, orange curve), the adaptive target (blue curve), the Expected Calibration Error (ECE, red curve) and error rate ($1 - accuracy$, black curve).

# G  Hyperparameter Sensitivity Analysis, Alternative Dynamic Uncertainty Penalties, and Ablation Study

To analyze our proposed Socrates method, we conducted a hyperparameter sensitivity analysis, an evaluation of alternative dynamic uncertainty penalties, and an ablation study on CIFAR-100 with VGG-16, using five random seeds.

## G.1  Hyperparameter Sensitivity

A hyperparameter sensitivity analysis was conducted on CIFAR-100 using VGG-16, based on the best tuning values: $\gamma = 2$ and $\alpha = 0.999$. Building on these settings, we explored the modularity factor and momentum factor across a wider range of values: $\gamma \in \{0, 1, 2, 3, 4\}$ and $\alpha \in \{0.8, 0.9, 0.99, 0.999, 1\}$. The results can be found in Tables 4 and 5, with the corresponding Pareto plots shown in Figure 10.

Table 4: Hyperparameter sensitivity analysis - $\gamma$ sensitivity. **ECE** values in a range of $[0, 1]$ and **accuracy** (%) scores (Acc) for the validation dataset including mean and standard deviation. Best results are highlighted in **bold** and second-best results are underlined. The models that perform best in terms of both accuracy and ECE, according to their Pareto plot positions, are shown in *italics* (epochs 150 and 300).

| Epoch | Metric | $\gamma$ sensitivity | | | | |
| | | 0 | 1 | 2 | 3 | 4 |
|---|---|---|---|---|---|---|
| 25 | Acc | **36.06 ± 0.64** | 34.56 ± 1.55 | 35.21 ± 2.01 | 35.22 ± 1.03 | 33.98 ± 0.58 |
| 25 | ECE | **2.37 ± 0.35** | 2.63 ± 1.06 | 3.10 ± 1.15 | 4.45 ± 1.48 | 5.00 ± 1.00 |
| 50 | Acc | **51.62 ± 0.53** | 50.95 ± 0.45 | 50.71 ± 0.33 | 49.48 ± 0.65 | 49.10 ± 0.68 |
| 50 | ECE | **2.10 ± 0.33** | 3.17 ± 0.99 | 5.64 ± 0.99 | 8.32 ± 1.42 | 10.60 ± 0.25 |
| 75 | Acc | **59.80 ± 0.73** | 58.48 ± 0.46 | 59.09 ± 0.91 | 57.65 ± 0.64 | 56.60 ± 0.50 |
| 75 | ECE | 3.40 ± 0.68 | **2.13 ± 0.65** | 6.29 ± 0.81 | 8.91 ± 0.78 | 11.67 ± 0.63 |
| 100 | Acc | **64.92 ± 0.36** | 64.47 ± 0.67 | 63.78 ± 0.54 | 62.99 ± 0.36 | 62.74 ± 0.53 |
| 100 | ECE | 5.19 ± 0.62 | **1.78 ± 0.30** | 5.06 ± 0.18 | 8.71 ± 0.51 | 12.83 ± 0.51 |
| 150 | Acc | **70.65 ± 0.21** | 69.78 ± 0.32 | *69.40 ± 0.44* | 69.23 ± 0.57 | 68.50 ± 0.40 |
| 150 | ECE | 8.83 ± 0.18 | 4.15 ± 0.59 | ***3.29 ± 0.21*** | 7.74 ± 0.25 | 11.32 ± 0.81 |
| 200 | Acc | **71.95 ± 0.17** | 71.45 ± 0.33 | 71.10 ± 0.14 | 70.79 ± 0.36 | 70.30 ± 0.27 |
| 200 | ECE | 11.24 ± 0.11 | 6.20 ± 0.63 | **3.30 ± 0.54** | 6.36 ± 0.46 | 10.09 ± 0.69 |
| 250 | Acc | **72.36 ± 0.23** | 71.99 ± 0.40 | 71.57 ± 0.18 | 71.32 ± 0.42 | 70.87 ± 0.48 |
| 250 | ECE | 11.77 ± 0.19 | 6.72 ± 0.33 | **3.29 ± 0.44** | 5.60 ± 0.52 | 9.62 ± 0.84 |
| 300 | Acc | **72.63 ± 0.17** | 72.11 ± 0.50 | *71.74 ± 0.34* | 71.45 ± 0.43 | 70.93 ± 0.32 |
| 300 | ECE | 11.85 ± 0.12 | 6.86 ± 0.42 | ***3.31 ± 0.42*** | 5.46 ± 0.46 | 9.37 ± 0.73 |

$\gamma$ is not sensitive to changes in terms of accuracy, meaning that a poor hyperparameter search for this value is not detrimental to accuracy performance. In contrast, in terms of confidence calibration, a good hyperparameter search can be favorable, leading to better-calibrated models. This component still holds importance, as seen when $\gamma = 0$, where the focal term becomes 1 and calibration deteriorates.

$\alpha$ is the hyperparameter that controls the weight between previous and current predictions and the initial target. A lower $\alpha$ places more importance on current predictions than on previous or initial target, which can be detrimental. In fact, in our hyperparameter study (Appendix F), the majority of the datasets preferred the 0.999 value, rather than 0.9. However, depends on the dataset, some models can work better in lower values as can be seen in Table 3.

Higher $\alpha$ values lead to higher accuracy and lower ECE. However, setting $\alpha = 1$ results in a version of Focal loss influenced by the extra unknown class and the adaptive target, and although it can still result in a well-calibrated model with good accuracy, its oscillatory calibration trend across epochs does not provide the stable performance that we seek. In contrast, $\alpha = 0.999$ offers a more consistent trend and the desirable behavior.

Table 5: Hyperparameter sensitivity analysis - $\alpha$ sensitivity. **ECE** values in a range of $[0, 1]$ and **accuracy** (%) scores (Acc) for the validation dataset including mean and standard deviation. Best results are highlighted in **bold** and second-best results are underlined. The models that perform best in terms of both accuracy and ECE, according to their Pareto plot positions, are shown in *italics* (epochs 150 and 300).

| Epoch | Metric | $\alpha$ sensitivity | | | | |
|---|---|---|---|---|---|---|
| | | 0.8 | 0.9 | 0.99 | 0.999 | 1 |
| 25 | Acc | $4.72 \pm 1.62$ | $17.30 \pm 0.76$ | $34.87 \pm 0.80$ | $\mathbf{35.21 \pm 2.01}$ | $\underline{34.97 \pm 0.72}$ |
| 25 | ECE | $15.36 \pm 2.08$ | $12.56 \pm 0.47$ | $5.48 \pm 1.82$ | $\underline{3.10 \pm 1.15}$ | $\mathbf{2.39 \pm 0.82}$ |
| 50 | Acc | $15.12 \pm 2.03$ | $27.98 \pm 2.17$ | $\underline{50.52 \pm 0.41}$ | $\mathbf{50.71 \pm 0.33}$ | $50.39 \pm 0.79$ |
| 50 | ECE | $16.69 \pm 1.49$ | $15.62 \pm 1.96$ | $10.20 \pm 0.98$ | $\underline{5.64 \pm 0.99}$ | $\mathbf{4.78 \pm 0.94}$ |
| 75 | Acc | $22.86 \pm 1.02$ | $38.80 \pm 0.92$ | $58.20 \pm 0.81$ | $\mathbf{59.09 \pm 0.91}$ | $\underline{58.50 \pm 0.51}$ |
| 75 | ECE | $16.29 \pm 1.51$ | $14.47 \pm 0.84$ | $9.83 \pm 0.92$ | $\underline{6.29 \pm 0.81}$ | $\mathbf{4.81 \pm 0.70}$ |
| 100 | Acc | $26.82 \pm 1.52$ | $45.02 \pm 0.94$ | $63.58 \pm 0.32$ | $\underline{63.78 \pm 0.54}$ | $\mathbf{64.01 \pm 0.31}$ |
| 100 | ECE | $16.48 \pm 1.35$ | $13.85 \pm 1.04$ | $10.12 \pm 0.68$ | $\underline{5.06 \pm 0.18}$ | $\mathbf{4.49 \pm 0.51}$ |
| 150 | Acc | $32.10 \pm 1.43$ | $52.88 \pm 0.52$ | $68.58 \pm 0.34$ | $\textit{\textbf{69.40} \pm \textit{0.44}}$ | $\underline{69.40 \pm 0.51}$ |
| 150 | ECE | $15.98 \pm 0.69$ | $11.32 \pm 0.49$ | $7.52 \pm 0.58$ | $\textit{3.29} \pm \textit{0.21}$ | $\underline{3.63 \pm 0.47}$ |
| 200 | Acc | $33.82 \pm 1.13$ | $56.65 \pm 0.64$ | $70.45 \pm 0.35$ | $\mathbf{71.10 \pm 0.14}$ | $\underline{71.02 \pm 0.44}$ |
| 200 | ECE | $15.42 \pm 1.09$ | $9.99 \pm 0.27$ | $5.77 \pm 0.56$ | $\mathbf{3.30 \pm 0.54}$ | $\underline{3.88 \pm 0.53}$ |
| 250 | Acc | $34.55 \pm 1.07$ | $57.71 \pm 0.56$ | $71.14 \pm 0.30$ | $\mathbf{71.57 \pm 0.18}$ | $\underline{71.57 \pm 0.32}$ |
| 250 | ECE | $14.67 \pm 0.96$ | $10.15 \pm 0.47$ | $4.93 \pm 0.15$ | $\underline{3.29 \pm 0.44}$ | $\mathbf{3.32 \pm 0.31}$ |
| 300 | Acc | $34.76 \pm 1.17$ | $58.14 \pm 0.31$ | $71.29 \pm 0.39$ | $\textit{\textbf{71.74} \pm \textit{0.34}}$ | $\underline{71.65 \pm 0.24}$ |
| 300 | ECE | $14.63 \pm 0.83$ | $10.01 \pm 0.11$ | $4.51 \pm 0.18$ | $\textit{3.31} \pm \textit{0.42}$ | $\underline{3.37 \pm 0.32}$ |

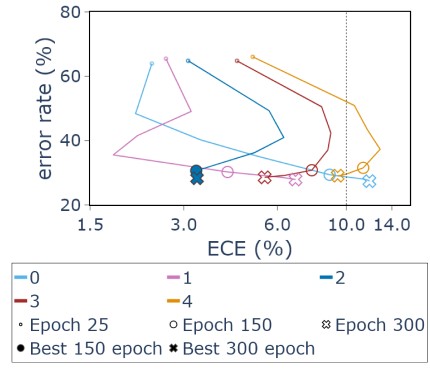

(a) $\gamma$ Pareto plot.

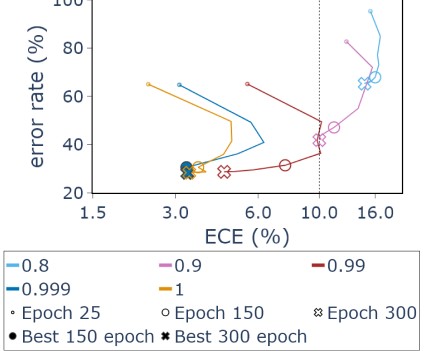

(b) $\alpha$ Pareto plot.

Figure 7: Error rate (1 - accuracy) versus Expected Calibration Error (ECE) across epochs for $\gamma$ (left) and $\alpha$ (right). Dotted lines indicate the threshold for acceptable calibration. Lower and leftward values indicate better performance. Lines are drawn every 25 epochs.

## G.2 Exploring Alternative Dynamic Uncertainty Penalties

Analyzing SAT, we found that the average confidence values of the unknown class are related to calibration. Specifically, when the model shows higher average confidence in the unknown class, ECE tends to change and increase, possibly due to incorrect confidence values for the ground truth classes. Examples for CIFAR-100 with VGG-16 and ResNet-110 architectures can be found in Figure 8.

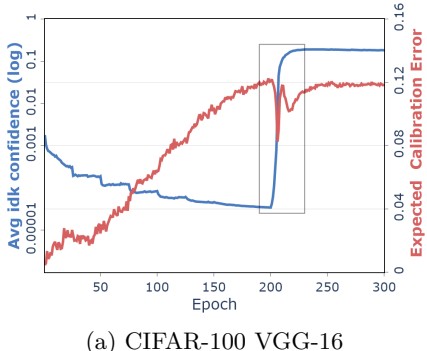

(a) CIFAR-100 VGG-16

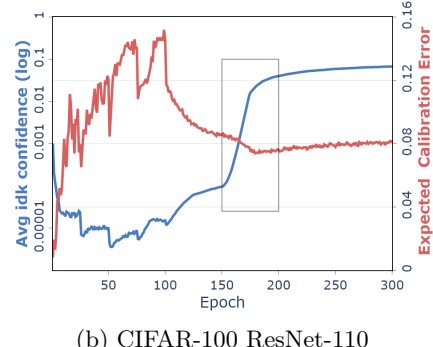

(b) CIFAR-100 ResNet-110

Figure 8: Curves depicting the average values of the unknown class confidences (left) and Expected Calibration Error (ECE) values (right) across epochs for CIFAR-100 with VGG-16 (a) and ResNet-110 (b), using the SAT method. Gray boxes highlight instability and miscalibration due to loss function change at epoch 200 (a) and 150 (b). It is evident that when confidence in the unknown class increases on average, ECE suddenly changes and increases as well.

This behavior inspired the development of Socrates, leading to the incorporation of the unknown class in the loss function, which enables confidence calibration during training. We argue that adding the unknown class and introducing $\beta$ in the loss function provides a key advantage for calibrated classifiers, allowing dynamic adjustment of penalization based on prediction confidence, improving both calibration and performance.

While the rationale behind the dynamic uncertainty penalty $\beta$ in Eq. 2 is clearly motivated in the main text as equal in Appendix D, we also explore alternative forms of dynamic uncertainty penalties:

1. $\beta$ **without ground truth class influence** ($\beta_{\backslash GT}$): Used in Socrates method. The maximum probability cannot correspond to the probability associated with the ground truth class. $\Rightarrow$ $\beta_{i,e} = \max_{\bar{y}_i \neq y_i} (\hat{p}_{i,\bar{y}_i,e}) - \hat{p}_{i,idk,e}$; s.t. $\beta \in [0,1]$.

2. $\beta$ **without ground truth class and unknown class influence** ($\beta_{\backslash GT,idk}$): The maximum probability cannot be the probability associated with the ground truth class or the probability associated with the extra unknown class. $\Rightarrow \beta_{i,e} = \max_{\bar{y}_i \neq (y_i \vee idk)} (\hat{p}_{i,\bar{y}_i,e}) - \hat{p}_{i,idk,e}$; s.t. $\beta \in [0,1]$.

3. $\beta$ **with fixed value** ($\beta_{fixed}$): We selected a random value of 0.25. $\Rightarrow$ $\beta_{i,e} = 0.25$ if $\hat{p}_{i,y_i,e} \leq \max_{\bar{y}_i \neq (y_i \vee idk)} (\hat{p}_{i,\bar{y}_i,e})$ else 0; s.t. $\beta \in [0,1]$.

4. **without $\beta$ influence** ($\backslash \beta$): $\Rightarrow \beta_{i,e} = 1; \beta \in [0,1]$.

Table 6 presents the results of this analysis, with the corresponding Pareto plot illustrated in Figure 9. The findings emphasize the necessity of a dynamic uncertainty penalty to achieve both improved calibration and accuracy, as shown by comparing $\backslash \beta$ with other types of $\beta$. It is noticeable that, in the case of $\backslash \beta$, the confidence calibration values increase throughout the epochs, worsening the calibration. Similarly to $\backslash \beta$, $\beta_{fixed}$ also exhibits increasing in confidence calibration values; however, the increase is less pronounced. Although $\beta_{\backslash GT,idk}$ achieves comparable results at epoch 300 than $\beta_{\backslash GT}$, its confidence calibration values oscillations across epochs are larger than those of our selected $\beta$.

We chose $\beta_{\backslash GT}$ for Socrates loss in accordance with the logic behind the method and and its consistent performance and steady trends across epochs. However, we do not dismiss $\beta_{\backslash GT,idk}$ and propose it as an alternative hyperparameter for further experimentation.

Table 6: Hyperparameter sensitivity analysis - $\beta$ sensitivity. **ECE** values in a range of $[0,1]$ and **accuracy** (%) scores (Acc) for the validation dataset including mean and standard deviation. Best results are highlighted in **bold** and second-best results are underlined. The models that perform best in terms of both accuracy and ECE, according to their Pareto plot positions, are shown in *italics* (epochs 150 and 300).

| Epoch | Metric | $\beta_{\backslash GT}$ | $\beta_{\backslash GT,idk}$ | $\beta_{fixed}$ | $\backslash\beta$ |
|---|---|---|---|---|---|
| | | | | $\beta$ sensitivity | |
| 25 | Acc | $\underline{35.21 \pm 2.01}$ | $\mathbf{35.72 \pm 0.95}$ | $34.95 \pm 0.81$ | $34.46 \pm 1.84$ |
| 25 | ECE | $3.10 \pm 1.15$ | $3.03 \pm 1.42$ | $\mathbf{2.09 \pm 0.86}$ | $\underline{2.33 \pm 1.40}$ |
| 50 | Acc | $\underline{50.71 \pm 0.33}$ | $\mathbf{51.21 \pm 0.27}$ | $50.39 \pm 0.33$ | $50.70 \pm 1.03$ |
| 50 | ECE | $5.64 \pm 0.99$ | $6.32 \pm 0.96$ | $\underline{5.01 \pm 0.45}$ | $\mathbf{4.90 \pm 1.23}$ |
| 75 | Acc | $\mathbf{59.09 \pm 0.91}$ | $58.56 \pm 0.58$ | $58.71 \pm 0.68$ | $\underline{58.73 \pm 0.47}$ |
| 75 | ECE | $6.29 \pm 0.81$ | $5.41 \pm 0.37$ | $\underline{4.77 \pm 0.70}$ | $\mathbf{4.58 \pm 0.56}$ |
| 100 | Acc | $63.78 \pm 0.54$ | $\underline{63.98 \pm 0.69}$ | $\mathbf{64.02 \pm 0.23}$ | $63.84 \pm 0.24$ |
| 100 | ECE | $5.06 \pm 0.18$ | $5.07 \pm 0.51$ | $\underline{3.93 \pm 0.42}$ | $\mathbf{3.20 \pm 0.48}$ |
| 150 | Acc | $69.40 \pm 0.44$ | $69.54 \pm 0.23$ | $\underline{69.67 \pm 0.19}$ | $\mathbf{\textit{69.75} \pm \textit{0.50}}$ |
| 150 | ECE | $3.29 \pm 0.21$ | $3.46 \pm 0.44$ | $\underline{3.14 \pm 0.20}$ | $\mathbf{\textit{2.83} \pm \textit{0.86}}$ |
| 200 | Acc | $71.10 \pm 0.14$ | $71.06 \pm 0.29$ | $\mathbf{71.41 \pm 0.41}$ | $\underline{71.38 \pm 0.47}$ |
| 200 | ECE | $\underline{3.30 \pm 0.54}$ | $\mathbf{3.02 \pm 0.30}$ | $3.35 \pm 0.25$ | $3.87 \pm 0.55$ |
| 250 | Acc | $71.57 \pm 0.18$ | $\underline{71.90 \pm 0.30}$ | $71.72 \pm 0.33$ | $\mathbf{71.92 \pm 0.27}$ |
| 250 | ECE | $\mathbf{3.29 \pm 0.44}$ | $\underline{3.39 \pm 0.53}$ | $3.97 \pm 0.30$ | $4.50 \pm 0.45$ |
| 300 | Acc | $\textit{71.74} \pm \textit{0.34}$ | $\mathbf{72.10 \pm 0.37}$ | $71.82 \pm 0.45$ | $\underline{71.98 \pm 0.38}$ |
| 300 | ECE | $\mathbf{\textit{3.31} \pm \textit{0.42}}$ | $\underline{3.45 \pm 0.50}$ | $4.05 \pm 0.39$ | $4.52 \pm 0.43$ |

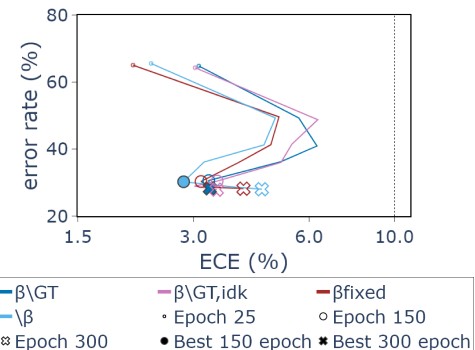

Figure 9: Error rate (1 - accuracy) versus Expected Calibration Error (ECE) across epochs for $\beta$. Dotted lines indicate the threshold for acceptable calibration. Lower and leftward values indicate better performance. Lines are drawn every 25 epochs.

### G.3 Ablation Study

An ablation study was conducted to evaluate the impact of each component of the Socrates loss on accuracy and calibration metrics. Key parts of the loss were systematically removed, while the final optimal hyperparameters for the main loss function were retained. Following prior research, these analyses were performed on CIFAR-100 with VGG-16 and ResNet-110, using 5 different seeds (1-5). The following functions were evaluated:

1. **Socrates Loss:** $\mathcal{L}_{\text{Soc}}(f)$

2. **Socrates without** $\beta \Rightarrow \mathcal{L}_{\text{Soc}\setminus\beta}(f) = -\frac{1}{n}\sum_{i=1}^{n}(1-\hat{p}_{i,y_i,e})^{\gamma}[t_{i,y_i,e}\log\hat{p}_{i,y_i,e} + (1-t_{i,y_i,e})\log\hat{p}_{i,idk,e}].$

3. **Socrates without focal term** $\Rightarrow \mathcal{L}_{\text{Soc}\setminus FT}(f) = -\frac{1}{n}\sum_{i=1}^{n}[t_{i,y_i,e}\log\hat{p}_{i,y_i,e} + \beta_{i,e}(1-t_{i,y_i,e})\log\hat{p}_{i,idk,e}].$

4. **Socrates without focal term in unknown component** $\Rightarrow$
   $\mathcal{L}_{\text{Soc}\setminus FT_{idk}}(f) = -\frac{1}{n}\sum_{i=1}^{n}(1-\hat{p}_{i,y_i,e})^{\gamma}t_{i,y_i,e}\log\hat{p}_{i,y_i,e} + \beta_{i,e}(1-t_{i,y_i,e})\log\hat{p}_{i,idk,e}.$

5. **Socrates without focal term in ground truth component** $\Rightarrow$
   $\mathcal{L}_{\text{Soc}\setminus FT_{gt}}(f) = -\frac{1}{n}\sum_{i=1}^{n}t_{i,y_i,e}\log\hat{p}_{i,y_i,e} + (1-\hat{p}_{i,y_i,e})^{\gamma}\beta_{i,e}(1-t_{i,y_i,e})\log\hat{p}_{i,idk,e}.$

6. **Socrates without focal term and** $\beta$ (equivalent to SAT) $\Rightarrow$
   $\mathcal{L}_{\text{Soc}\setminus FT,\beta}(f) = -\frac{1}{n}\sum_{i=1}^{n}t_{i,y_i,e}\log\hat{p}_{i,y_i,e} + (1-t_{i,y_i,e})\log\hat{p}_{i,idk,e}].$

7. **Socrates without focal term in unknown component and** $\beta \Rightarrow$
   $\mathcal{L}_{\text{Soc}\setminus FT_{idk},\beta}(f) = -\frac{1}{n}\sum_{i=1}^{n}(1-\hat{p}_{i,y_i,e})^{\gamma}t_{i,y_i,e}\log\hat{p}_{i,y_i,e} + (1-t_{i,y_i,e})\log\hat{p}_{i,idk,e}.$

8. **Socrates without focal term in ground truth component and** $\beta \Rightarrow$
   $\mathcal{L}_{\text{Soc}\setminus FT_{gt},\beta}(f) = -\frac{1}{n}\sum_{i=1}^{n}t_{i,y_i,e}\log\hat{p}_{i,y_i,e} + (1-\hat{p}_{i,y_i,e})^{\gamma}(1-t_{i,y_i,e})\log\hat{p}_{i,idk,e}].$

9. **Socrates without adaptive target** (equivalent to Focal loss) $\Rightarrow$
   $\mathcal{L}_{\text{Soc}\setminus t_a}(f) = -\frac{1}{n}\sum_{i=1}^{n}(1-\hat{p}_{i,y_i,e})^{\gamma}t_{i,y_i,e}\log\hat{p}_{i,y_i,e};$ with $t_{i,y_i,e} = y_i.$

10. **Socrates without adaptive target and focal term** (equivalent to CE) $\Rightarrow$
    $\mathcal{L}_{\text{Soc}\setminus t_a,FT}(f) = -\frac{1}{n}\sum_{i=1}^{n}t_{i,y_i,e}\log\hat{p}_{i,y_i,e};$ with $t_{i,y_i,e} = y_i.$

The results for CIFAR-100 (VGG-16) can be found in Table 7 and for CIFAR-100 (ResNet-110) in Table 8.

Our proposed loss function ($\mathcal{L}_{\text{Soc}}$) ranked in the top-1 once the models reach competitive accuracy, while the variant without the focal term in the unknown component ($\mathcal{L}_{\text{Soc}\setminus FT_{idk}}$) and the variant without the dynamic uncertainty penalty ($\mathcal{L}_{\text{Soc}\setminus\beta}$) rank second and third respectively. This does not imply that these components are irrelevant for confidence calibration: removing both ($\mathcal{L}_{\text{Soc}\setminus FT_{idk},\beta}$) leads to a substantial increase in the confidence calibration error. However, it remains unclear which component contributes more strongly to this degradation.

Removing the focal term from the ground truth component ($\mathcal{L}_{\text{Soc}\setminus FT}$, $\mathcal{L}_{\text{Soc}\setminus FT_{gt}}$, $\mathcal{L}_{\text{Soc}\setminus FT,\beta}$, $\mathcal{L}_{\text{Soc}\setminus FT_{gt},\beta}$, and $\mathcal{L}_{\text{Soc}\setminus t_a,FT}$) produces consistently suboptimal results, with all models reaching the worst calibration values. We attribute this deterioration primarily to the removal of the focal term from the ground truth component: removing the focal term only from the unknown component has a comparatively minor effect, as seen by contrasting $\mathcal{L}_{\text{Soc}\setminus FT}$ and $\mathcal{L}_{\text{Soc}\setminus FT_{gt}}$. This highlights the importance of including at least the focal term in the ground truth component together with the dynamic uncertainty penalty.

Eliminating the adaptive target ($\mathcal{L}_{\mathrm{Soc}\backslash t_a}$) also degrades confidence calibration, and this effect becomes more severe when the focal component is removed as well ($\mathcal{L}_{\mathrm{Soc}\backslash t_a,FT}$), which produces the highest confidence calibration errors.

When jointly considering accuracy and confidence calibration, a key observation emerges: $\mathcal{L}_{\mathrm{Soc}\backslash FT,\beta}$ exhibits some of the worst confidence calibration values for both models and causes a substantial accuracy drop for VGG-16. This underscores the effectiveness and necessity of these components and reveals a novel relationship between the dynamic uncertainty penalty and confidence calibration. **Thus, $\beta$ alone could potentially be considered a standalone calibration component for future research.**

The analysis revealed that each component is essential and significantly contributes to the final confidence calibration and classification result.

Table 7: Ablation Study for CIFAR-100 with VGG-16. **ECE** values and **accuracy** scores (Acc) in % for the validation dataset including mean. Standard deviations are omitted due to space constraints and to facilitate clearer comparison, as they remained below 2% for accuracy and 0.2% for ECE. The results in **bold** represent the best outcomes and those in underlined the second-best. The models that perform best in terms of both accuracy and ECE, according to their Pareto plot positions, are shown in *italics* (epochs 150 and 300).

| Epoch | Metric | $\mathcal{L}_{\mathrm{Soc}}$ | $\mathcal{L}_{\mathrm{Soc}\backslash\beta}$ | $\mathcal{L}_{\mathrm{Soc}\backslash FT}$ | $\mathcal{L}_{\mathrm{Soc}\backslash FT_{idk}}$ | $\mathcal{L}_{\mathrm{Soc}\backslash FT_{gt}}$ | $\mathcal{L}_{\mathrm{Soc}\backslash FT,\beta}$ | $\mathcal{L}_{\mathrm{Soc}\backslash FT_{idk},\beta}$ | $\mathcal{L}_{\mathrm{Soc}\backslash FT_{gt},\beta}$ | $\mathcal{L}_{\mathrm{Soc}\backslash t_a}$ | $\mathcal{L}_{\mathrm{Soc}\backslash t_a,FT}$ |
|---|---|---|---|---|---|---|---|---|---|---|---|
| | | | | | | Loss functions | | | | | |
| 25 | Acc | 35.21 | 34.46 | **36.06** | 35.46 | 35.42 | 35.21 | 34.99 | 35.98 | 34.74 | 35.64 |
| 25 | ECE | 3.10 | 2.33 | 2.37 | 3.09 | 2.51 | 2.67 | 3.19 | **2.19** | 2.93 | 2.54 |
| 50 | Acc | 50.71 | 50.70 | 51.62 | 50.32 | 51.77 | 51.04 | 50.89 | **52.10** | 50.55 | 51.71 |
| 50 | ECE | 5.64 | 4.90 | **2.10** | 5.59 | 2.25 | 2.45 | 6.49 | 2.89 | 2.13 | 2.65 |
| 75 | Acc | 59.09 | 58.73 | **59.80** | 58.63 | 59.62 | 59.07 | 58.90 | 60.10 | 59.52 | 59.60 |
| 75 | ECE | 6.29 | 4.58 | 3.40 | 5.36 | 3.46 | 4.80 | 6.55 | 4.51 | **2.08** | 4.39 |
| 100 | Acc | 63.78 | 63.84 | 64.92 | 63.83 | 65.05 | 64.76 | 63.62 | 65.00 | 64.33 | **65.06** |
| 100 | ECE | 5.06 | 3.20 | 5.19 | 4.71 | 4.94 | 6.36 | 6.74 | 6.76 | **1.96** | 6.42 |
| 150 | Acc | 69.40 | *69.75* | 70.65 | 70.01 | 70.27 | 69.98 | 69.63 | **70.66** | 69.71 | 70.16 |
| 150 | ECE | 3.29 | *2.83* | 8.83 | 3.80 | 8.92 | 10.30 | 7.43 | 9.77 | 4.58 | 10.25 |
| 200 | Acc | 71.10 | 71.38 | 71.95 | 71.38 | **71.99** | 71.80 | 71.48 | 71.91 | 71.47 | 71.94 |
| 200 | ECE | **3.30** | 3.87 | 11.24 | 3.92 | 10.86 | 12.06 | 8.09 | 12.07 | 6.64 | 11.97 |
| 250 | Acc | 71.57 | 71.92 | 72.36 | 71.93 | **72.40** | 66.06 | 71.74 | 72.34 | 72.00 | 72.21 |
| 250 | ECE | **3.29** | 4.50 | 11.77 | 3.98 | 11.54 | 11.80 | 8.67 | 12.53 | 7.16 | 12.87 |
| 300 | Acc | *71.74* | 71.98 | **72.63** | 72.07 | 72.38 | 66.53 | 71.74 | 72.48 | 72.08 | 72.36 |
| 300 | ECE | *3.31* | 4.52 | 11.85 | 3.90 | 11.83 | 11.87 | 8.93 | 12.73 | 7.43 | 13.01 |

Table 8: Ablation Study for CIFAR-100 with ResNet-110. **ECE** values and **accuracy** scores (Acc) in % for the validation dataset including mean. Standard deviations are omitted due to space constraints and to facilitate clearer comparison, as they remained below 2% for accuracy and 0.2% for ECE. The results in **bold** represent the best outcomes and those in underlined the second-best. The models that perform best in terms of both accuracy and ECE, according to their Pareto plot positions, are shown in *italics* (epochs 150 and 300).

| Epoch | Metric | $\mathcal{L}_{\mathrm{Soc}}$ | $\mathcal{L}_{\mathrm{Soc}\backslash\beta}$ | $\mathcal{L}_{\mathrm{Soc}\backslash FT}$ | $\mathcal{L}_{\mathrm{Soc}\backslash FT_{idk}}$ | $\mathcal{L}_{\mathrm{Soc}\backslash FT_{gt}}$ | $\mathcal{L}_{\mathrm{Soc}\backslash FT,\beta}$ | $\mathcal{L}_{\mathrm{Soc}\backslash FT_{idk},\beta}$ | $\mathcal{L}_{\mathrm{Soc}\backslash FT_{gt},\beta}$ | $\mathcal{L}_{\mathrm{Soc}\backslash t_a}$ | $\mathcal{L}_{\mathrm{Soc}\backslash t_a,FT}$ |
|---|---|---|---|---|---|---|---|---|---|---|---|
| | | | | | | Loss functions | | | | | |
| 25 | Acc | 50.66 | 50.99 | 52.33 | 50.89 | 52.31 | **53.00** | 50.48 | 51.54 | 50.35 | 48.43 |
| 25 | ECE | 6.53 | 5.69 | 8.53 | 5.44 | 8.99 | 7.96 | 4.64 | 8.76 | **2.51** | 11.87 |
| 50 | Acc | 61.04 | 60.92 | 61.81 | 61.90 | **61.92** | 60.78 | 60.96 | 60.78 | 57.98 | 60.37 |
| 50 | ECE | 6.61 | 7.25 | 10.31 | 6.05 | 10.97 | 12.07 | 3.97 | 11.24 | **3.21** | 10.26 |
| 75 | Acc | 65.30 | 65.08 | 65.93 | 65.61 | **66.58** | 66.01 | 65.00 | 65.66 | 64.18 | 66.06 |
| 75 | ECE | 9.48 | 9.95 | 13.94 | 9.26 | 12.91 | 14.03 | 5.24 | 14.53 | **3.09** | 12.97 |
| 100 | Acc | 68.90 | 69.26 | **69.91** | 69.08 | 69.66 | 68.82 | 68.79 | 68.89 | 67.71 | 69.26 |
| 100 | ECE | 8.88 | 10.41 | 13.88 | 8.89 | 14.01 | 14.56 | 3.67 | 15.20 | **2.41** | 14.26 |
| 150 | Acc | *78.18* | 77.64 | 78.01 | **78.40** | 78.37 | 77.81 | 76.03 | 77.68 | 77.60 | 77.88 |
| 150 | ECE | *2.56* | 5.23 | 8.93 | 2.85 | 8.41 | 8.59 | 6.06 | 9.50 | 6.60 | 9.89 |
| 200 | Acc | 78.57 | 77.93 | 78.00 | 78.71 | 78.85 | 77.21 | 76.03 | 78.07 | 77.78 | **79.00** |
| 200 | ECE | **2.69** | 4.94 | 9.23 | 2.96 | 8.23 | 7.60 | 6.54 | 9.29 | 7.43 | 9.34 |
| 250 | Acc | 78.47 | 78.00 | 78.18 | 78.58 | 78.74 | 76.58 | 76.23 | 78.10 | 77.75 | **78.82** |
| 250 | ECE | **2.65** | 5.10 | 9.19 | 2.79 | 8.34 | 7.78 | 6.36 | 9.37 | 7.47 | 9.59 |
| 300 | Acc | *78.39* | 78.00 | 78.16 | 78.46 | **78.91** | 76.18 | 76.14 | 78.20 | 77.77 | 78.83 |
| 300 | ECE | *2.64* | 5.14 | 9.27 | 3.01 | 8.29 | 8.06 | 6.39 | 9.23 | 7.62 | 9.51 |

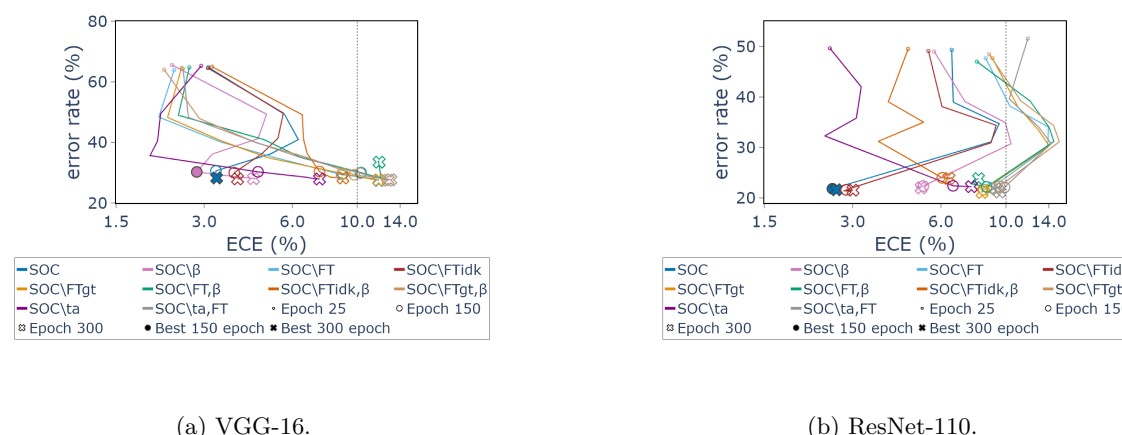

(a) VGG-16.

(b) ResNet-110.

Figure 10: Error rate (1 - accuracy) versus Expected Calibration Error (ECE) across epochs for VGG-16 (left) and ResNet-110 (right). Dotted lines indicate the threshold for acceptable calibration. Lower and leftward values indicate better performance. Lines are drawn every 25 epochs.

# H  Unknown Class Behaviour During Training

The dynamic uncertainty penalty penalizes the model for failing to recognize its own uncertainty by penalizing when any probability not associated with the ground truth class exceeds the probability associated with the unknown class. This penalty directs the model to explicitly account for both the confidence associated with the ground truth class and the confidence associated with the unknown class; augmenting the unknown confidence if the model does not recognize its own uncertainty. Figure 11 illustrates this effect, showing that the average confidence for the ground truth class increases over the epochs, as does the average confidence for the unknown class. In general, the ground truth confidences are higher than the ones associated with the unknown class.

When the model predicts correctly (i.e., the predicted class matches the ground truth), the average confidence associated with the ground truth class is higher than the ones when the model does not predict correctly. Conversely, when the model fails to predict the ground truth, the average confidence in the unknown class is higher than when the prediction is correct. This empirical behavior demonstrates that the model allocates higher unknown-class confidence precisely when it fails.

Moreover, because the model is penalized for failing to recognize its own uncertainty, the confidence assigned to the unknown class gradually increases over training epochs. As a result, the model increasingly classify instances as unknown class when uncertainty is high, as reflected in the frequency of the unknown class being predicted as top-1 (Figure 11, bottom). Over the course of training, the dynamic uncertainty penalty acts as a regularizer while also providing the model with an explicit uncertainty mechanism through the additional unknown class, which represents uncertainty and complements the regularization effect.

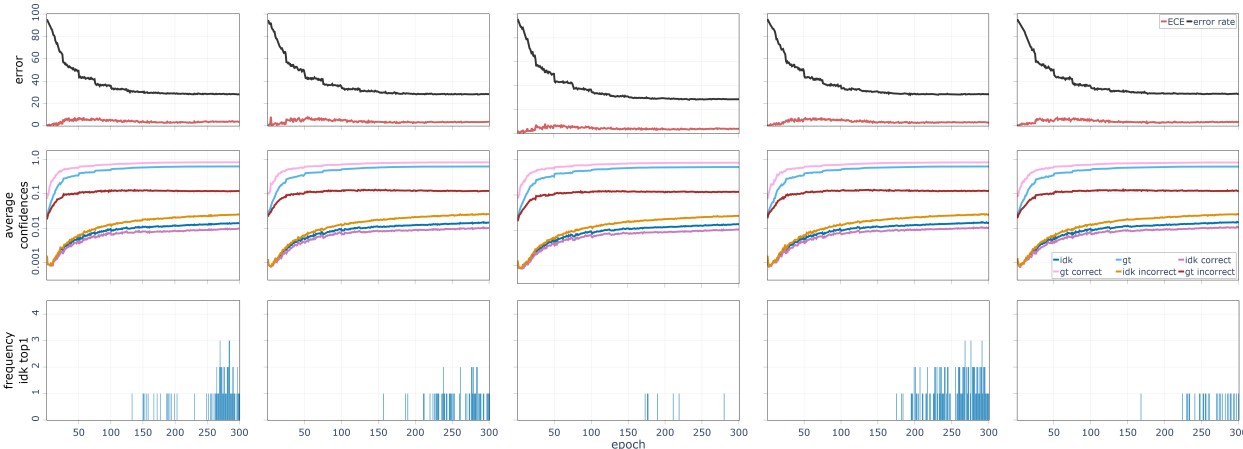

Figure 11: Curves illustrating the evolution of the Socrates method on CIFAR-100 with VGG-16 across multiple random seeds, evaluated on the validation set. The top plot shows the Expected Calibration Error (ECE, red curve) and error rate ($1 - accuracy$, black curve) across epochs. The middle plot illustrates the average confidence values across epochs for the unknown class (denoted as idk, dark-blue curve) and the ground-truth class (denoted as gt, light-blue curve), including: 1) the idk (purple curve) and gt (pink curve) confidences when the model's predictions are correct, and 2) the idk (orange curve) and gt (brown curve) confidences when the model's predictions are incorrect. The bottom plot shows the frequency with which the idk class is selected as the top-1 prediction across epochs.

# I Validation Set Performance

As a reference for future research and to illustrate the performance achieved on the validation set, we report the validation results in Table 9.

Table 9: Final validation set performance at epoch 300 for standard training and epoch 50 for transfer learning (TL). Metrics reported: accuracy, ECE, AdaptiveECE, and Classwise-ECE. Poor performance in SAT with Food-101, CCL-SC and MC with SVHN is attributed to premature convergence. Best results are highlighted in **bold**, and second-best are underlined.

| | Metric | Socrates | SAT | CCL-SC | Focal | FLSD | Brier | MC |
|---|---|---|---|---|---|---|---|---|
| CIFAR-10 VGG-16 | Acc | **88.80 ± 0.21** | 88.37 ± 0.29 | 88.29 ± 0.35 | 88.53 ± 0.17 | 88.51 ± 0.52 | 88.24 ± 0.33 | 88.02 ± 0.22 |
| | ECE | **4.70 ± 0.41** | 8.91 ± 0.33 | 9.22 ± 0.28 | 6.12 ± 0.26 | 6.03 ± 0.46 | 6.81 ± 0.31 | 8.98 ± 0.29 |
| | AdaECE | **6.20 ± 0.50** | 8.98 ± 0.31 | 9.17 ± 0.24 | 6.38 ± 0.32 | 6.47 ± 0.50 | 7.01 ± 0.39 | 8.97 ± 0.30 |
| | CW-ECE | **1.34 ± 0.04** | 1.40 ± 0.04 | 1.56 ± 0.04 | 1.48 ± 0.03 | 1.47 ± 0.06 | 1.55 ± 0.07 | 1.51 ± 0.04 |
| SVHN VGG-16 | Acc | 96.51 ± 0.08 | 96.50 ± 0.08 | 19.95 ± 0.00 | 65.75 ± 41.82 | 81.09 ± 34.18 | **96.73 ± 0.13** | 35.25 ± 34.23 |
| | ECE | 2.31 ± 0.13 | 0.95 ± 0.08 | **0.93 ± 0.28** | 4.65 ± 2.72 | 2.43 ± 0.67 | 1.37 ± 0.18 | 3.01 ± 0.95 |
| | AdaECE | 2.14 ± 0.11 | 1.09 ± 0.08 | **0.93 ± 0.28** | 4.52 ± 2.60 | 2.43 ± 0.69 | 2.43 ± 0.18 | 3.02 ± 0.92 |
| | CW-ECE | **1.11 ± 0.01** | 1.15 ± 0.02 | 6.44 ± 0.50 | 3.19 ± 2.60 | 2.11 ± 2.20 | 1.12 ± 0.01 | 5.05 ± 2.20 |
| Food-101 ResNet-34 | Acc | 73.72 ± 0.35 | 12.61 ± 28.19 | 69.21 ± 4.76 | 67.83 ± 11.96 | 70.29 ± 6.83 | **73.85 ± 0.28** | 71.08 ± 0.47 |
| | ECE | **1.49 ± 0.23** | 81.55 ± 38.04 | 8.23 ± 0.90 | 1.60 ± 0.86 | 4.95 ± 0.15 | 4.35 ± 0.24 | 8.62 ± 0.36 |
| | AdaECE | **1.48 ± 0.22** | 81.55 ± 38.04 | 8.17 ± 0.90 | 1.67 ± 0.90 | 4.93 ± 0.12 | 4.28 ± 0.26 | 8.59 ± 0.35 |
| | CW-ECE | **0.29 ± 0.01** | 1.60 ± 0.57 | 0.34 ± 0.05 | 0.36 ± 0.14 | 0.33 ± 0.08 | 0.30 ± 0.01 | 0.32 ± 0.01 |
| CIFAR-100 VGG-16 | Acc | 71.74 ± 0.34 | 66.52 ± 0.22 | **73.06 ± 0.60** | 72.07 ± 0.47 | 70.73 ± 0.48 | 53.75 ± 1.74 | 68.61 ± 0.45 |
| | ECE | **3.31 ± 0.42** | 11.87 ± 0.21 | 11.85 ± 0.82 | 7.44 ± 0.23 | 5.60 ± 0.42 | 6.94 ± 0.31 | 7.80 ± 0.39 |
| | AdaECE | **3.31 ± 0.47** | 12.28 ± 0.18 | 11.74 ± 0.76 | 7.44 ± 0.26 | 5.66 ± 0.45 | 7.66 ± 0.27 | 7.75 ± 0.40 |
| | CW-ECE | 0.32 ± 0.01 | 0.51 ± 0.01 | 0.34 ± 0.01 | **0.31 ± 0.00** | 0.34 ± 0.00 | 0.52 ± 0.02 | 0.33 ± 0.00 |
| CIFAR-100 ResNet-110 | Acc | 78.39 ± 0.35 | 76.17 ± 0.83 | **78.73 ± 0.68** | 77.76 ± 0.59 | 77.81 ± 0.55 | 75.09 ± 0.69 | 75.45 ± 0.47 |
| | ECE | **2.64 ± 0.48** | 8.07 ± 0.31 | 9.98 ± 0.66 | 7.62 ± 2.03 | 3.73 ± 1.25 | 4.10 ± 0.28 | 12.99 ± 0.39 |
| | AdaECE | **2.42 ± 0.48** | 8.07 ± 0.31 | 9.91 ± 0.61 | 7.54 ± 2.16 | 3.60 ± 1.24 | 4.03 ± 0.16 | 12.98 ± 0.39 |
| | CW-ECE | **0.28 ± 0.00** | 0.34 ± 0.01 | 0.28 ± 0.01 | 0.32 ± 0.00 | 0.28 ± 0.01 | 0.32 ± 0.01 | 0.31 ± 0.01 |
| CIFAR-100 ViT | Acc | 49.07 ± 3.52 | 43.16 ± 0.90 | 47.79 ± 5.34 | 48.42 ± 5.11 | **49.17 ± 3.75** | 42.86 ± 0.67 | 45.12 ± 9.37 |
| | ECE | 3.97 ± 0.98 | 21.49 ± 0.54 | 11.62 ± 5.42 | 8.73 ± 3.49 | 10.80 ± 2.69 | **1.67 ± 0.59** | 6.28 ± 2.03 |
| | AdaECE | 4.07 ± 0.92 | 21.48 ± 0.53 | 11.60 ± 5.48 | 8.72 ± 3.49 | 10.80 ± 2.69 | **1.69 ± 0.28** | 6.27 ± 2.04 |
| | CW-ECE | **0.49 ± 0.03** | 0.75 ± 0.02 | 0.55 ± 0.10 | 0.52 ± 0.09 | 0.52 ± 0.06 | 0.57 ± 0.01 | 0.56 ± 0.13 |
| CIFAR-100 ViT TL | Acc | 92.16 ± 0.14 | 71.49 ± 19.19 | **92.91 ± 0.25** | 69.44 ± 20.62 | 77.44 ± 18.50 | 58.78 ± 21.73 | 63.16 ± 26.61 |
| | ECE | 5.82 ± 0.14 | 1.83 ± 0.48 | **0.96 ± 0.23** | 6.96 ± 1.33 | 2.05 ± 0.53 | 7.22 ± 3.18 | 2.05 ± 0.85 |
| | AdaECE | 5.81 ± 0.15 | 1.81 ± 0.53 | **0.92 ± 0.19** | 6.93 ± 1.37 | 1.97 ± 0.58 | 7.44 ± 2.99 | 2.04 ± 0.78 |
| | CW-ECE | 0.19 ± 0.01 | 0.36 ± 0.17 | **0.16 ± 0.00** | 0.40 ± 0.18 | 0.30 ± 0.17 | 0.53 ± 0.19 | 0.45 ± 0.25 |

## J   Test Set Performance - Pareto Plots

Pareto plots were used to identify the best model for each architecture and dataset at epoch 300 on the test set, with error rate plotted against ECE.. In cases of ambiguity (i.e., two models occupying the same Pareto position), ties were resolved using Classwise-ECE, a quantitative metric reported in the main results. Figure 12 presents the Pareto plots for all baselines versus Socrates, showing that Socrates achieves the top-1 performance in most cases, except for SVHN. Comparisons between Socrates and CE, including post-hoc results, are illustrated in Figure 13, where Socrates and its post-hoc variants generally outperform CE. For CIFAR-100 with ViT, CE and its post-hoc versions achieve higher Pareto positions, although this should be interpreted in light of the relatively poor accuracy. Transfer learning results for ViT are shown in Figure 14, where CCL-SC excels; among the post-hoc methods, Socrates using Vector Scaling (VS) demonstrates superior performance.

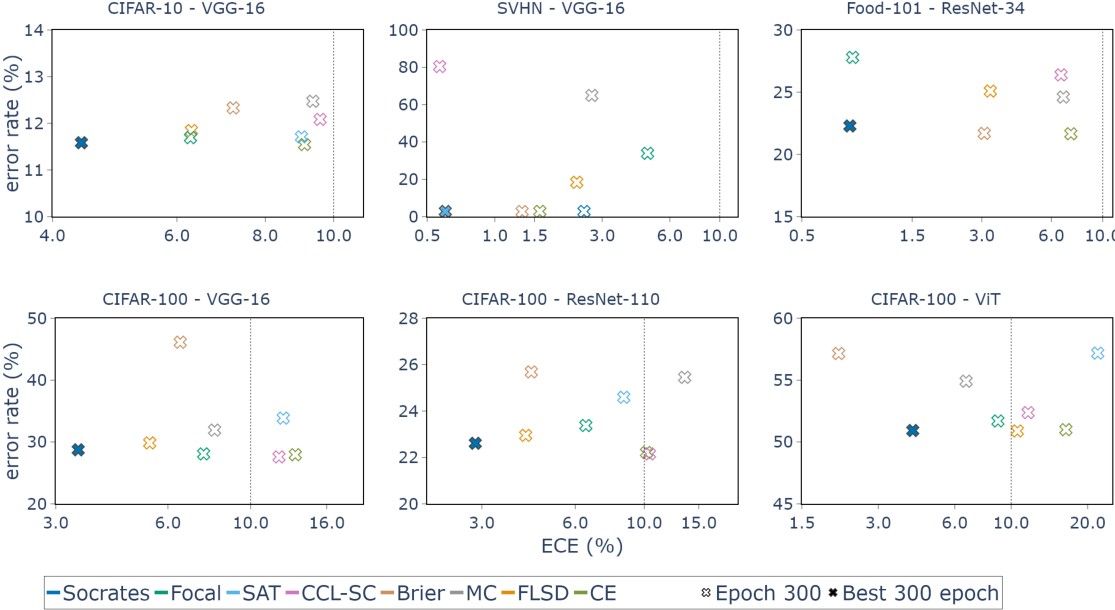

Figure 12: Error rate (1 - accuracy) versus Expected Calibration Error (ECE) at the last epoch for different dataset–architectures on the test set. Dotted lines indicate the threshold for acceptable calibration. Lower and leftward values indicate better performance. SAT is excluded from Food-101 due to premature convergence.

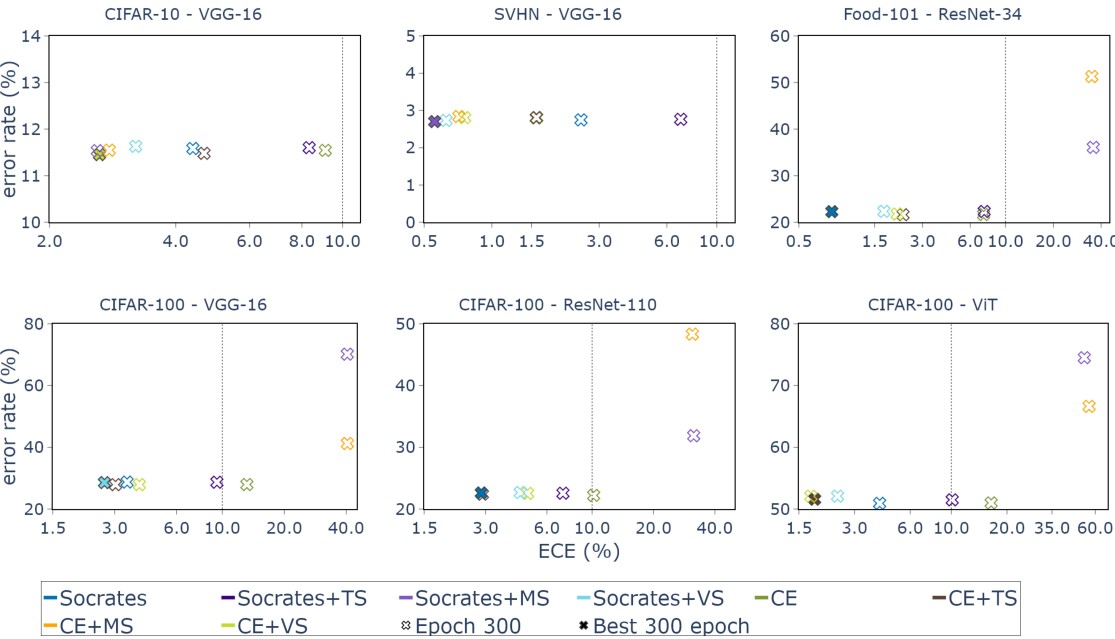

Figure 13: Post-hoc results (Socrates and CE baselines). Error rate (1 - accuracy) versus Expected Calibration Error (ECE) at the last epoch for different dataset–architectures on the test set. Dotted lines indicate the threshold for acceptable calibration. Lower and leftward values indicate better performance. In the SVHN plot, the CE result is not visible because it is overlapped by CE+TS result.

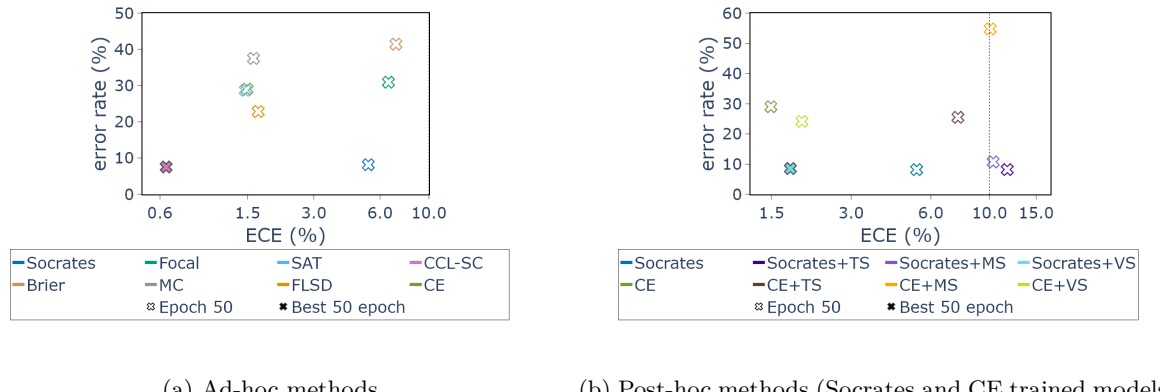

(a) Ad-hoc methods        (b) Post-hoc methods (Socrates and CE trained models)

Figure 14: Pareto plot at the last epoch for the CIFAR-100 test set using ViT with Transfer Learning (TL) comparing ad-hoc methods (a) and post-hoc methods (b). CE is included in the comparison with post-hoc methods.

