# OpenReview forum: "Socrates Loss: Unifying Confidence Calibration and Classification by Leveraging the Unknown"
_TMLR — Accepted by TMLR_

### Review · Reviewer_yZiN · 2025-12-20

**Summary Of Contributions:**

This paper introduces Socrates loss, a novel approach to improve deep neural network calibration by incorporating an auxiliary class and modifying the Kullback-Leibler (KL) divergence loss. The method encourages the network to predict the auxiliary class in uncertain scenarios by adding a dedicated regularization term. The authors provide theoretical insights demonstrating that Socrates loss serves as a lower bound of the standard KL divergence loss. Empirical comparisons with non-architectural baseline methods highlight its advantages, particularly in terms of accuracy and Expected Calibration Error.

### Strengths

- The paper is well-written and clearly structured.
- The proposed method is simple, non-intrusive, and computationally efficient, making it easy to implement.
- The theoretical analysis enhances understanding of the method’s behavior.
- The experimental evaluation is thorough, spanning four datasets, four metrics, and three architectures. The epoch-loss plots are notably informative.
- The authors openly provide anonymized PyTorch implementation code, which is a commendable effort for reproducibility.

### Weaknesses

- I am not totally convinced by the contribution offered by General Calibration Error, as contribution appears to be a weighted average of existing metrics without clear additional insight or justification.
- A comparison with deep ensembles—a strong, widely adopted baseline for calibration—is missing, which would have strengthened the empirical validation.

**Audience:**

Yes

**Audience Explanation:**

This paper tackles confidence calibration, a well-known and critical challenge in deep learning. The proposed method is likely to interest the TMLR audience due to its non-intrusive nature, ease of implementation, and low computational overhead, making it both practical and widely applicable.

**Claims And Evidence:**

Yes

**Claims Explanation:**

The claims made in the submission are supported by accurate, convincing, and clear evidence. The benefits of the proposed Socrates loss are thoroughly explained, both conceptually and empirically. The experimental setup is comprehensive, covering multiple datasets, architectures, and metrics. While the method introduces hyperparameters, these are extensively explored in the appendix. Additionally, the theoretical proofs appear sound based on my review.

**Requested Changes:**

I have no major requested changes—the paper is already strong in its current form. However, I suggest the following minor improvements to further enhance its impact:

- Include results for no calibration procedure to quantify the trade-off between accuracy and calibration performance.

- Add a comparison with deep ensembles, which remain one of the most widely used calibration methods.

- Investigate how Socrates loss behaves in a fine-tuning setup:
  - Can the method effectively calibrate pre-trained models?
  - How many epochs are required for convergence in this scenario?
  - Does fine-tuning yield results equivalent to training from scratch?

---

> ### Author Response · Authors · 2026-01-27
> **Response to Reviewer yZiN**
>
> ### 1. **GCE Metric.** Shared Response (All authors).
> We have considered all reviewers’ suggestions and agree that the GCE metric warrants further thought and analysis. Confidence calibration metrics are inherently complex, and designing a single metric that combines calibration and classification remains an open research problem. Therefore, we have decided to explore this in greater depth in future work (analyzing existing metrics thoroughly and proposing an improved metric as a separate study) and  maintain focus on the main contribution: Socrates, in this work.
>
> Accordingly, we have removed the GCE metric from the manuscript (and the appendix) and have instead selected the best models using Pareto plots, as recommended by Reviewer @qYad; plotting error rate versus ECE and, in case of ambiguity (i.e., two models in the same Pareto position), resolve the tie using Classwise-ECE. These plots have also been added to the appendix. We believe this approach is the most consistent with the rest of the manuscript and allows a coherent evaluation across datasets and models.  This change improves the overall consistency of the manuscript (noting that the GCE metric was previously used only in Subsection 4.2 – Last epoch performance).
>
> We acknowledge that while visual assessment via Pareto plots is useful, there is a clear lack of quantitative metrics that offer a robust model selection criterion consistent with qualitative evaluation, while avoiding an excessive emphasis on accuracy (as in the Brier Score); that is precisely the issue we aimed to address. This constitutes an important direction for future work.
>
> ### 2. **Minor: Include results for no calibration procedure to quantify the trade-off between accuracy and calibration performance.**
>
> We thank the reviewer for this observation. We have added results for CE and CE combined with post-hoc calibration methods. These results can be found in Table 2, and are also plotted in the Figures 2-5 and discussed along Section 4.2. We initially did not include CE because it typically produces overconfident models; however, we agree that including it is a good practice for a fair comparison, especially given that it is the most widely used loss.
>
> ### 3. **Minor: Deep ensembles.**
>
> We acknowledge that deep ensembles are a widely used approach for improving calibration. However, incorporating deep ensembles in this context raises important methodological questions that require careful consideration, such as which loss functions should be used to train the ensemble. For instance, should all models be trained solely with CE, or should ensembles be constructed separately for each method, or even by combining different losses? Addressing these questions in this context would require a significantly more complex experimental design and a level of analysis that goes beyond the scope of the current work. We believe that such an investigation deserves dedicated attention and would be better suited for future work. Accordingly, we chose not to include deep ensembles in our experimental setup due to the substantial computational cost of training all possible scenarios.
>
> ### 4. **Minor: Fine-tuning setup.**
>
> Thank you for these insightful questions. We refer the reviewer to Subsection 4.2 (Transfer Learning). In summary, in our experiments we ran Socrates alongside the other methods for 50 epochs. We observed that, in comparison, Socrates trained with a stable trend, achieving higher accuracy (91.83) and lower calibration error (ECE 5.30) than a model trained from scratch (accuracy 49.07, ECE 4.10).

---

### Review · Reviewer_k9s2 · 2026-01-14

**Summary Of Contributions:**

The paper proposes Socrates Loss, a single training objective intended to jointly improve classification accuracy and confidence calibration by adding an auxiliary “unknown / idk” class and coupling it with (i) a focal-style weighting, (ii) a self-adaptive target, and (iii) a dynamic uncertainty penalty. Experiments on SVHN, CIFAR-10/100, and Food-101 across several architectures show slightly improved calibration metrics and an accuracy–calibration tradeoff.

**Audience:**

Yes

**Audience Explanation:**

Confidence calibration is an important and broadly relevant topic, and the paper’s goal of a single objective that improves both accuracy and calibration should appeal to researchers in trustworthy ML and practical deployment. That said, the current experimental scope limits how general the conclusions feel; with stronger validation (more datasets/architectures), the findings would be more compelling to a wider portion of the audience.

**Broader Impact Concerns:**

The work is a methodological contribution on confidence calibration, so there are no obvious direct ethical red flags.

**Claims And Evidence:**

No

**Claims Explanation:**

Overall, the claims are only partially supported by clear and convincing evidence.

First, while the motivation for most loss components is clear, the role of the focal term is not. Beyond empirical gains, I would like a clearer explanation of why this term specifically helps close the accuracy–calibration gap. In prior work, focal loss is mainly used to improve classification performance by emphasizing hard examples; its connection to confidence calibration is not obvious. The ablations in the appendix did not convincingly demonstrate that this term is necessary rather than simply beneficial in some settings. Moreover, there are presentation issues: Tables 8 and 9 appear to have incorrect bold/italic markings, which makes it harder to interpret the intended takeaways.

Second, since the method introduces an “idk” class, it would be important to report how this behaves during training. In particular, what is the trend of the model’s probability/confidence assigned to “idk” over epochs? Showing how often “idk” is predicted over time would clarify whether it is functioning as a meaningful uncertainty mechanism or primarily acting as a regularizer.

Third, the experimental validation feels limited in scope. The number/diversity of datasets and model architectures are relatively small for claims about general effectiveness, and additional settings (more architectures, larger-scale datasets) would strengthen the empirical case. I would suggest adding results on ImageNet and testing modern architectures such as Swin Transformer and ConvNeXt for a better analysis.

Finally, I am not convinced by the definition of GCE. Averaging (1−accuracy) with three calibration metrics mixes quantities with different scales and semantics, and the equal weighting implicitly prioritizes calibration error over accuracy without a clear justification. Either the weighting should be explicitly justified (and potentially rebalanced), or the paper should report a composite score based only on calibration metrics.

**Requested Changes:**

Critical adjustments:

1. Justify the focal term’s role in calibration: Provide a clear conceptual explanation for why the focal component should improve calibration (not only accuracy), ideally connecting it to confidence dynamics (e.g., margin behavior, overconfidence on hard examples, effect on logit scaling).

2. Fix presentation errors (bold/italic markings) in Appendix Tables 8 and 9.

3. Analyze the “idk” class behavior over training: Report the trend of “idk” over epochs: mean predicted probability, frequency of selecting “idk,” and how it correlates with misclassification or low-margin examples. Clarify whether “idk” is intended as an uncertainty mechanism or purely as a regularizer, and evaluate accordingly.

4. Reconsider the General Calibration Error (GCE) or justify its weighting: Either justify/rebalance the weighting, or report a score based only on calibration metrics.

Strengthening adjustments:

1. Broaden empirical validation (datasets + architectures): Add larger-scale benchmarks and modern architectures to test generality, e.g., ImageNet and models like Swin Transformer and ConvNeXt.

---

> ### Author Response · Authors · 2026-01-27
> **Response to Reviewer k9s2 - Critical Adjustments**
>
> ### 1. **Major: Justify the focal term’s role in calibration.**
> We thank the reviewer for the suggestion, we clarified and strengthened the motivation for this component in Subsection 3.2. In particular, we added the following conceptual explanation:
>
> “The focal term controls the dominance of easy-to-classify instances by down-weighting their contribution to the loss, thereby shifting the effective gradient focus toward hard-to-classify instances. In the absence of this term (e.g., CE), easy-to-classify instances contribute to the loss and gradient even at high confidence levels, increasing ground truth logits and inflating classification margins, i.e., increasing the separation between the ground truth confidence and competing class confidences. While this margin inflation may marginally improve accuracy, it induces overconfident predictions, as there is no mechanism to regulate logit growth once high confidence is reached, widening the accuracy–calibration gap. By reducing the influence of easy instances, the focal term stabilizes margin growth and implicitly constrains logit scaling, encouraging predicted confidence to better align with instance difficulty. This mechanism mitigates overconfidence and explains why the focal term improves calibration in addition to classification performance.”
>
> This effect can also be observed in Tables 8 and 9 (e.g., from epoch 150) and in the corresponding Pareto plots (Figure 10), where removing the focal term, particularly from the ground truth component where its impact is bigger (functions 3, 5, and 6), leads to increased ECE values. These results show that the focal term is not only beneficial but also essential for stable calibration.
>
> ### 2. **Major: Table formatting**
> To ensure consistency with the evaluation procedure used throughout the manuscript, we have updated the formatting in the Appendix, including Table 8 and 9. Specifically, we have modified the bold/underlined according to the best and second best per metric, and indicated the best model (according to its position on the Pareto plot) in italic. Previously, the best model was indicated in bold based on ECE and accuracy combined; we have now changed this for consistency.
>
> ### 3. **Major: Analyze the “idk” class behavior over training.**
>
> We have carried out this analysis in Appendix H, demonstrating how the unknown class serves as part of the model's uncertainty mechanism. To summarize, the average confidence for both the ground truth class and the unknown class increases over the training epochs, with the latter increase due to the dynamic uncertainty penalty, which penalizes the model for failing to recognize its own uncertainty. When predictions are correct, the average confidence assigned to the ground truth class is higher than when predictions are incorrect, whereas the average confidence assigned to the unknown class is higher when the model fails than when it is correct. Furthermore, the frequency with which the unknown class is predicted as top-1 gradually increases over the epochs, reflecting its role both as part of the model’s uncertainty mechanism and as a contributor to the regularization effect of the dynamic penalty.
>
> ### 4. **Major: GCE Metric.** Shared Response (All authors).
> We have considered all reviewers’ suggestions and agree that the GCE metric warrants further thought and analysis. Confidence calibration metrics are inherently complex, and designing a single metric that combines calibration and classification remains an open research problem. Therefore, we have decided to explore this in greater depth in future work (analyzing existing metrics thoroughly and proposing an improved metric as a separate study) and  maintain focus on the main contribution: Socrates, in this work.
>
> Accordingly, we have removed the GCE metric from the manuscript (and the appendix) and have instead selected the best models using Pareto plots, as recommended by Reviewer @qYad; plotting error rate versus ECE and, in case of ambiguity (i.e., two models in the same Pareto position), resolve the tie using Classwise-ECE. These plots have also been added to the appendix. We believe this approach is the most consistent with the rest of the manuscript and allows a coherent evaluation across datasets and models.  This change improves the overall consistency of the manuscript (noting that the GCE metric was previously used only in Subsection 4.2 – Last epoch performance).
>
> We acknowledge that while visual assessment via Pareto plots is useful, there is a clear lack of quantitative metrics that offer a robust model selection criterion consistent with qualitative evaluation, while avoiding an excessive emphasis on accuracy (as in the Brier Score); that is precisely the issue we aimed to address. This constitutes an important direction for future work.

---

> ### Author Response · Authors · 2026-01-27
> **Response to Reviewer k9s2 - Strengthening Adjustment**
>
> ### 1. **Minor: Strengthening adjustment.**
> We agree that evaluating on additional datasets and architectures is always valuable. However, we have designed our experiments to balance coverage and resource constraints, while still addressing a wide range of scenarios: from simpler benchmark datasets to more challenging real-world dataset such as Food-101. Our experiments already include multiple architectures, including CNNs, ResNet, ViT, and transfer learning setups. We fully acknowledge the potential to explore more datasets, architectures, and applications of Socrates (e.g., in OOD detection, selective classification, and other reliability tasks) in future work. Nevertheless, we believe that the current experimental setup provides a meaningful evaluation for the manuscript, and we would like to focus on the critical suggestions provided.

---

### Review · Reviewer_qYad · 2026-01-14

**Summary Of Contributions:**

Summary of Contributions

This submissions proposes Socrates Loss, a unified training loss for deep neural network classification that tries to jointly improve calibration and classification performance, but maintaining stable optimization dynamics. This is because the authors claim (and show empirically) that there is a stability–performance trade-off in ad-hoc calibration losses: approaches with time-varying or two-stages objectives (meaning scheduled or multi-loss training) can achieve strong accuracy but often paying a price of unstable training dynamics and degrading calibration, whereas single-loss methods tend to be more stable but sometimes underperform in accuracy. This Socrates loss is designed to preserve the optimization stability of single-loss training while achieving competitive accuracy and calibration. The method introduces an auxiliary unknown class whose predicted probability intends to modulate the loss through a dynamic uncertainty penalty, allowing uncertainty awareness to be integrated end-to-end instead of through post-hoc calibration or scheduled multi-loss training.

The paper gives an extensive empirical evaluation across multiple datasets, architectures (including several CNNs and vision transformers), and training regimes, with some attention to training dynamics, convergence behavior, and calibration evolution over epochs. Ablation studies analyze the role played by the main loss components and hyperparameters, and comparisons are conducted against a wide set of ad-hoc and post-hoc calibration strategies.

Relative to the earlier ICLR submission of this work, the current version expands the experimental scope, improves clarity and organization, adds ablations and sensitivity analyses, and addresses most of the concerns previous reviewrs had regarding empirical validation and presentation. In summary, this work is mostly empirical, and contributes a comparative analysis of calibration-aware training strategies, together with a concrete loss designed to mitigate some observed instability issues.

**Additional Comments:**

None

**Audience:**

Yes

**Audience Explanation:**

The findings of this paper are likely to be of interest to researchers working on model calibration, reliability, and uncertainty-aware learning, and also those using deep classifiers in settings where confidence estimates and training stability are relevant. The empirical analysis of training dynamics and failure modes of existing ad-hoc calibration methods provides useful (and probably novel) insight beyond final-epoch performance comparisons, and the proposed loss offers a concrete alternative for integrating calibration objectives directly into training. The comparative evaluation and discussion of stability issues in calibration-aware training are informative and relevant to the TMLR audience.

**Broader Impact Concerns:**

No concern.

**Claims And Evidence:**

Yes

**Claims Explanation:**

The main empirical claims of the paper are generally supported by experimental evidence. In particular, the authors demonstrate that several existing ad-hoc calibration approaches with scheduled or time-varying objectives can exhibit unstable training behavior, including sudden changes in loss and calibration metrics, and in some cases premature convergence or degradation of calibration. These observations are documented consistently across multiple datasets, architectures, and random seeds, and are supported by training-dynamics plots and per-epoch calibration analyses.

The Socrates Loss is shown to have smoother optimization dynamics while achieving competitive final accuracy and calibration compared to a range of ad-hoc and post-hoc baselines. Ablation studies support the role of the main loss components and seem to indicate that this behavior is not driven by a single design choice or hyperparameter setting. The expanded experimental coverage relative to the earlier ICLR submission, including additional architectures and transfer-learning scenarios, supports the authors' conclusions comprehensively.

That said, some claims, specifically the ones related to achieving a "state-of-the-art balance" between accuracy and calibration, rely on composite evaluation metrics the design of which raises some questions and would benefit from further clarification, please see below. In any case, the core empirical observations regarding training stability, calibration behavior, and comparative performance are well described and supported by the reported results.

**Requested Changes:**

1) **Major: GCE metric**

The proposed General Calibration Error (GCE) raises methodological concerns that should be addressed before acceptance.

* GCE combines metrics that live on different natural scales (ECE, AdaECE, CW-ECE, and 1-accuracy) using equal weighting, without normalization or decision-theoretic justification.
* In particular, ECE and AdaECE are highly correlated estimators of the same underlying quantity  (only binning strategy changes here). This becomes quite obvious if one looks at Tables 1 and 2, where they result in nearly identical rankings: whenever a method is bold or underlined in ECE, it is also bold or underlined in AdaECE. Including both effectively double-counts the same calibration signal in an un-weighted average of four terms. This does not sound like a good idea.
* Related to this, despite the emphasis placed on GCE (with a section of its own). neither GCE nor AdaECE or CW-ECE appear to be used in hyperparameter selection or ablation analysis (see Appendix), where evaluation is instead carried out mostly in terms of accuracy and ECE alone. This implicitly treats ECE as a sufficient calibration metric, which further questions the justification for introducing and emphasizing a more complex composite metric. In other words, if the authors themselves do not use GCE for picking hyperparameters, that speaks poorly for their conviction on this being a good metric in this context.
* Indeed, Appendix C motivates the use of multiple metrics in general, but does not analyze redundancy or correlation among them, nor justify the specific choice and weighting used in GCE.

As a result, GCE is difficult to interpret as a principled evaluation criterion, and its use to support "state-of-the-art balance" claims is not really convincing, I believe. More generally, if the goal is to summarize performance across heterogeneous metrics, a ranking-based aggregation would be more appropriate than averaging raw metric values, given that the metrics involved live on different natural scales. Rank aggregation avoids arbitrary scale effects and better reflects the ordinal nature of most comparative claims made in the paper.

However, even under a ranking-based approach, there's still the need to avoid double-counting highly correlated metrics. In particular, including both ECE and AdaECE-even as ranks would still amount to overweighting the same calibration signal. A more principled aggregation, maybe, could be to restrict attention to a minimal and conceptually distinct set of metrics, most naturally accuracy and ECE (with CW-ECE included only if classwise calibration is a specific concern, this I guess might be dataset dependent?). Alternatively, the authors may wish to avoid composite metrics altogether and rely on Pareto-style or dominance-based analysis, which would be completely justified.

2) **Major: Missing basic baselines: Cross-Entropy and CE + Temperature Scaling**

The experimental evaluation omits results for a plain cross-entropy (CE) baseline, which is a widely used reference point in the calibration literature. CE is a proper scoring function, and should in principle exhibit decent calibration, so let us see how well it fares.

On the other hand, post-hoc calibration methods are discussed in the paper, but note that TS (as well as MS and VS) is only evaluated as a post-hoc correction applied to the proposed Socrates model (Table 2), and does not appear as a standalone baseline (in combination with CE). As a result, it is difficult to assess how much of the reported performance gains stem from the proposed loss itself, as opposed to improvements that could already be obtained by applying simple and well-established calibration techniques to a standard CE-trained model.

Including CE and CE+TS would provide an essential reference for interpreting the empirical results and would help contextualize the advantages of Socrates relative to the simplest common sense alternative. If the authors believe such baselines are not appropriate for this study, then they should probably justify this explicitly.

3) **Minor: Strength of the "stability–performance trade-off" claims**

The paper provides some empirical evidence that certain calibration-aware methods with scheduled or time-varying objectives can exhibit unstable training dynamics, and that the proposed Socrates Loss behaves more smoothly across datasets and architectures. However, framing this observation as a resolution of a stability–performance trade-off is a bit too much. Given that the evidence is purely observational and empirical rather than a formal characterization of this a trade-off, it would be more accurate to slightly soften the language in a few places (e.g., from "resolves" to "mitigates" or "empirically alleviates"). This would better align the strength of the claims with the nature of the evidence presented. This is probalby more of a personal comment, not a hill I would die on to be honest.

---

> ### Author Response · Authors · 2026-01-27
> **Response to Reviewer qYad**
>
> ### 1. **Major: GCE Metric.** Shared Response (All authors).
> We have considered all reviewers’ suggestions and agree that the GCE metric warrants further thought and analysis. Confidence calibration metrics are inherently complex, and designing a single metric that combines calibration and classification remains an open research problem. Therefore, we have decided to explore this in greater depth in future work (analyzing existing metrics thoroughly and proposing an improved metric as a separate study) and  maintain focus on the main contribution: Socrates, in this work.
>
> Accordingly, we have removed the GCE metric from the manuscript (and the appendix) and have instead selected the best models using Pareto plots, as recommended by Reviewer @qYad; plotting error rate versus ECE and, in case of ambiguity (i.e., two models in the same Pareto position), resolve the tie using Classwise-ECE. These plots have also been added to the appendix. We believe this approach is the most consistent with the rest of the manuscript and allows a coherent evaluation across datasets and models.  This change improves the overall consistency of the manuscript (noting that the GCE metric was previously used only in Subsection 4.2 – Last epoch performance).
>
> We acknowledge that while visual assessment via Pareto plots is useful, there is a clear lack of quantitative metrics that offer a robust model selection criterion consistent with qualitative evaluation, while avoiding an excessive emphasis on accuracy (as in the Brier Score); that is precisely the issue we aimed to address. This constitutes an important direction for future work.
>
> ### 2. **Major: Missing basic baselines: Cross-Entropy and CE + Temperature Scaling.**
> We thank the reviewer for this observation. We have added results for CE and CE combined with post-hoc calibration methods. These results can be found in Table 2, and are also plotted in the Figures 2-5 and discussed in Section 4.2. We initially did not include CE because it typically produces overconfident models; however, we agree that including it is a good practice for a fair comparison, especially given that it is the most widely used loss. Please also refer to our response to Reviewer @k9s2 for a related discussion on overconfidence and CE.
>
> #### **Major: Justify the focal term’s role in calibration** (reviewer @k92):
>
> We thank the reviewer for the suggestion, we clarified and strengthened the motivation for this component in Subsection 3.2. In particular, we added the following conceptual explanation:
>
> “The focal term controls the dominance of easy-to-classify instances by down-weighting their contribution to the loss, thereby shifting the effective gradient focus toward hard-to-classify instances. In the absence of this term (e.g., CE), easy-to-classify instances contribute to the loss and gradient even at high confidence levels, increasing ground truth logits and inflating classification margins, i.e., increasing the separation between the ground truth confidence and competing class confidences. While this margin inflation may marginally improve accuracy, it induces overconfident predictions, as there is no mechanism to regulate logit growth once high confidence is reached, widening the accuracy–calibration gap. By reducing the influence of easy instances, the focal term stabilizes margin growth and implicitly constrains logit scaling, encouraging predicted confidence to better align with instance difficulty. This mechanism mitigates overconfidence and explains why the focal term improves calibration in addition to classification performance.”
>
> This effect can also be observed in Tables 8 and 9 (e.g., from epoch 150), where removing the focal term, particularly from the ground truth component where its impact is bigger (functions 3, 5, and 6), leads to increased ECE values. These results show that the focal term is not only beneficial but also essential for stable calibration.
>
> ### 3. **Minor: Strength of the "stability–performance trade-off" claims.**
> We acknowledge the comment and have made minor changes to soften the wording, while preserving the clarity of the manuscript.

---

> ### Comment · Action_Editor_NUk5 · 2026-02-17
> **Fina recommendation**
>
> Dear reviewer,
>
> please read the author responses to the raised concerns and submit your final recommendation.
>
> Best, your AE.

---

> > ### Comment · Reviewer_qYad · 2026-02-19
> > **Final Recommendation**
> >
> > Hello,
> >
> > I wrote my final recommendation in a text file, but when I came here to upload it, I realized I cannot find a place where to do this. I tried editing my previous review, but no luck. Please find below the text I wrote, do let me know if I need to upload it anywere else, sorry for the clumsiness.
> >
> > It seems to me that the authors have addressed all my main concerns. They have removed the "polemic" GCE metric, and instead adopted a Pareto-based model selection (makes sense), added the missing CE and CE + Temperature Scaling baselines for fair comparison, and clarified the role of the focal term in calibration, all this supported by new ablation results. They have also analyzed the behavior of the "unknown" class and improved overall clarity throughout the paper. With these revisions, I think the manuscript is a bit stronger and significantly more consistent. I am satisfied with the responses, and happy to support acceptance.

---

### Author Response · Authors · 2026-01-27
**Revised Manuscript**

Dear Reviewers,

We sincerely thank you for your careful reading of our manuscript and for the constructive comments, which have helped us improve both the manuscript and the clarity of the proposed method for future readers.

In the revised manuscript, all changes are highlighted in blue. We provide detailed responses to each comment below.

Thank you for your time and consideration.

Sincerely,

Authors

---

### Decision · Action_Editor_NUk5 · 2026-03-03

**Recommendation:** Accept as is

**Additional Comments:**

N/A

**Audience:**

Yes

**Audience Explanation:**

Yes, researchers in uncertainty calibration may be interested in the findings from this paper.

**Claims And Evidence:**

Yes

**Claims Explanation:**

This paper investigates the stability-performance trade-off of training-time calibration, and propose a unified loss function that allows the model to be optimized for both objectives simultaneously, i.e., classification and confidence calibration.

I am recommending the paper’s acceptance. During the review and discussion phase, concerns were raised about the proposed GCE metric, missing baselines, and several clarifications, such as the role of the focal term in calibration. The authors provided detailed responses during the rebuttal, including additional experiments and clearer explanations that directly addressed these points. With the added modifications, I believe that this submission clearly supports the claims made.